# POINT-UQ: AN UNCERTAINTY-QUANTIFICATION PARADIGM FOR POINT CLOUD FEW-SHOT CLASS INCREMENTAL LEARNING

**Xiangqi Li**[1,2], **Libo Huang**[1*], **Jiarui Zhao**[2], **Weilun Feng**[1,2], **Chuanguang Yang**[1],
**Zhulin An**[1*], **Yongjun Xu**[1]

[1] State Key Laboratory of AI Safety, Institute of Computing Technology,
 Chinese Academy of Sciences
[2] University of Chinese Academy of Sciences

## ABSTRACT

3D few-shot class-incremental learning (3D FSCIL) requires effectively integrating novel classes from limited samples while preserving base-class knowledge, without succumbing to catastrophic forgetting the learned knowledge or overfitting the novel ones. Current 3D FSCIL approaches predominantly focus on fine-tuning feature representations yet retain static decision boundaries. This leads to a critical trade-off: excessive adaptation to new samples tends to erase previously learned knowledge, while insufficient adaptation hinders novel-class recognition. We argue that the key to effective incremental learning lies not only in feature enhancement but also in adaptive decision-making. To this end, we introduce **Point-UQ**, an incremental training-free paradigm for 3D **point** clouds based on **u**ncertainty **q**uantification, which shifts the focus from feature tuning to dynamic decision optimization. Point-UQ comprises two co-designed modules: *Attention-driven Adaptive Enhancement (AAE)* and *Uncertainty-quantification Decision Decoupling (UDD)*. The former module fuses multi-scale features into calibrated representations, where prediction entropy serves as a reliable measure of per-sample epistemic uncertainty while preserving original feature semantics. Building on AAE-derived calibrated entropy, the UDD module dynamically arbitrates between semantic classifiers and geometric prototypes—enabling robust base-class knowledge retention and accurate novel-class recognition in 3D FSCIL without retraining. Extensive experiments on ModelNet, ShapeNet, ScanObjectNN, and CO3D demonstrate that our approach outperforms state-of-the-art methods by $4\%$ in average accuracy, setting a new standard for robust 3D incremental learning.

## 1 INTRODUCTION

3D point clouds serve as the critical data sources for robotic navigation, augmented reality, and autonomous driving (Resani et al., 2025; Miao et al., 2024; Sun et al., 2024b; An et al., 2025b). Although the rapid development of 3D point cloud analysis techniques, how to use these data in the open-world environment is still stubbed (Xiao et al., 2024). One typical scenario is 3D few-shot class incremental learning (FSCIL), where point cloud models must first train on abundant synthetic base-class data, then incrementally adapt using minimal real-scanned samples per novel class (Li et al., 2025; Ahmadi et al., 2024; Tan & Xiang, 2024). This dual challenge induces catastrophic forgetting of base knowledge, insufficient discriminative power for novel classes, and cross-domain distribution shifts between synthetic base and open-world incremental data (Chowdhury et al., 2022; Cheraghian et al., 2025).

To address these challenges, Microshape (Chowdhury et al., 2022) pioneered a unified feature representation to bridge domain gaps, offering a foundational framework for multi-task compatibility.

---

*This work was supported by the National Natural Science Foundation of China (No.62476264 and No.62406312). Corresponding authors: Libo Huang (huanglibo@ict.ac.cn) and Zhulin An (anzhulin@ict.ac.cn).

Building on this, subsequent studies enhanced category separation through label replay strategies, demonstrating improved stability in sequential learning scenarios (Tan & Xiang, 2024). Recent advancements have explored multimodal alignment techniques, exemplified by C3PR (Cheraghian et al., 2025), which mapped 3D point clouds to 2D depth images to harness CLIP's cross-modal knowledge. While achieving promising semantic transfer, this projection-based paradigm inevitably sacrifices inherent 3D geometric fidelity (Wang et al., 2022). In a parallel development, the FoundationModel approach directly leverages large-scale 3D vision-language models, aiming to strengthen semantic representation through multimodal grounding (Ahmadi et al., 2024).

Despite their differences, a common thread among these methods is the heavy reliance on meticulous feature fine-tuning strategies to enhance feature discriminability, while retaining static decision boundaries throughout incremental stages (as illustrated in Fig.1(a)). This paradigm faces a fundamental dilemma: insufficient fine-tuning weakens the model's ability to discriminate novel classes and exacerbates overfitting to base classes, while aggressive fine-tuning causes overfitting to the limited novel samples and accelerates catastrophic forgetting (Lahoud et al., 2022; Tang et al., 2024). Moreover, repeated fine-tuning in incremental phases not only raises training costs but also increases the risk of forgetting previously learned knowledge. Notably, current approaches focus predominantly on refining feature representations through fine-tuning, while overlooking the potential of improving the decision-making process itself.

We argue that effective incremental learning depends not only on discriminative features but also on flexible decision-making. This motivates a shift in perspective: rather than continually chasing feature refinement through repetitive fine-tuning, a more efficient path lies in dynamically allocating existing knowledge via adaptive decision logic to balance base-class retention and novel-class adaptation, especially when feature expressive power is inherently bounded. Specifically, we demonstrate that uncertainty quantification provides a powerful basis for designing dynamic reasoning paths, which markedly improve adaptability and robustness in incremental scenarios with minimal parameter overhead. As illustrated in Fig. 1(b), this decision-centric strategy supports efficient learning without repeated fine-tuning, thereby avoiding the high training costs and forgetting risks inherent in conventional methods.

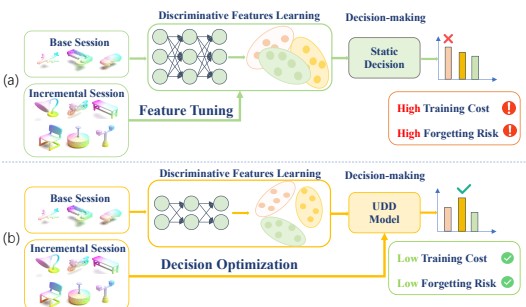

Figure 1: Motivation of Point-UQ: shift the focus from feature tuning to decision optimization. **(a) Traditional paradigm** relies heavily on discriminative feature learning via continuous fine-tuning. It leads to high training overhead and escalating risk of model catastrophic forgetting as incremental sessions proceed. Its static decision mechanism limits adaptation to new classes without disrupting previously learned knowledge. **(b) Point-UQ** focuses on optimizing the decision process based on existing features. By introducing the Uncertainty-quantification Decision Decoupling (UDD) module, the model eliminates the need for fine-tuning throughout incremental learning—significantly reducing computational cost and avoiding complexity growth. The UDD module dynamically guides the decision-making process based on uncertainty quantification, enabling flexible yet robust adaptation to novel classes without retraining.

We instantiate this perspective with **Point-UQ**, a novel **point** cloud decision optimization framework based on **u**ncertainty-**q**uantification. It consists of two synergistically designed modules: The Attention-driven Adaptive Enhancement (AAE) module first integrates multi-scale features during base training to produce calibrated representations. Crucially, it leverages prediction entropy as a reliable estimate of per-sample epistemic uncertainty, thereby preserving original semantic information while supplying essential uncertainty cues for subsequent decision stages. Building on these uncertainty signals, the Uncertainty-quantification Decision Decoupling (UDD) module dynamically arbitrates between semantic classifiers and geometric prototypes. For high-uncertainty samples, UDD shifts reliance toward more robust geometric similarities, while low-uncertainty predictions retain semantic-based decisions. This mechanism enables the model to retain knowledge of old classes and accurately recognize new class samples without the need for retraining. Extensive experiments on ModelNet, ShapeNet, ScanObjectNN, and CO3D demonstrate Point-UQ's superiority, with con-

sistent improvements in average accuracy, harmonic mean, and forgetting rate over state-of-the-art baselines (Wu et al., 2015; Chang et al., 2015; Uy et al., 2019; Reizenstein et al., 2021; Chowdhury et al., 2022; Tan & Xiang, 2024; Cheraghian et al., 2025; Ahmadi et al., 2024). In summary, our work makes the following key contributions:

1. We propose the Attention-driven Adaptive Enhanced Module, which integrates multi-scale features for calibrated representations, emphasizes cross-scale discriminative patterns, and leverages prediction entropy as reliable epistemic uncertainty. This supplies essential cues for subsequent decision stages, laying the groundwork for adaptive decision-making.

2. We design the Uncertainty-quantification Decision Decoupling Module, which introduces entropy-driven adaptive weighting using AAE-derived entropy. This strategy balances semantic classification and geometric prototype matching: prioritizing geometric cues for uncertain high-entropy samples while maintaining semantic boundaries for confident predictions, resolving old-new class conflicts.

3. We establish Point-UQ by synergistically integrating AAE and UDD—an incremental training-free 3D FSCIL paradigm. It reframes the problem by shifting focus from feature tuning to decision optimization, effectively balancing base class knowledge retention with rapid novel class adaptation. Extensive experiments on multiple 3D point cloud benchmarks show it consistently outperforms existing approaches across various evaluation metrics.

## 2 RELATED WORK

### 2.1 CONTINUAL LEARNING ON POINT CLOUD

Early research in point cloud processing laid the foundation by adapting 2D vision principles to the 3D domain (Li et al., 2025; Qi et al., 2017a;b; Wang et al., 2019; Zhao et al., 2021; Xu et al., 2018; Thomas et al., 2019; Huang et al., 2024a). For instance, PointNet (Qi et al., 2017a) introduced permutation-invariant feature learning through multi-layer perceptrons and global max-pooling, while PointNet++ (Qi et al., 2017b) improved local geometric modeling via hierarchical point sampling and grouping. With advancements in 3D modeling techniques, increasing attention has shifted to more complex tasks such as few-shot segmentation (An et al., 2024b;a; 2025a) and continual learning in 3D vision (Resani & Nasihatkon, 2024). Chowdhury *et al.* (Chowdhury et al., 2021) incorporated knowledge distillation with semantic word embeddings to alleviate catastrophic forgetting. The geometric attention mechanism in I3DOL (Dong et al., 2021) and fairness constraints in InOR-Net (Dong et al., 2023) enhanced structural preservation. CL3D (Resani et al., 2025) and RCR (Zamorski et al., 2023) further improved performance through spectral clustering and memory-efficient compression strategies. MIRACLE3D (Resani & Nasihatkon, 2024) constructs compact shape prototypes for each class, capturing both average structures and key variations, significantly reducing memory overhead without storing raw point clouds. These methods collectively address the core challenges of geometric structure preservation, memory efficiency, and incremental adaptability in continual learning on point clouds.

### 2.2 3D FEW-SHOT CLASS INCREMENTAL LEARNING

While multimodal learning, particularly involving images and point clouds, has driven significant progress in 3D vision (Li et al., 2025; Yang et al., 2022b; 2024; Feng et al., 2025; Huang et al., 2024b; Feng et al., 2024), applying these advances to continuous learning scenarios remains challenging. Specifically, 3D few-shot class incremental learning extends the framework of traditional FSCIL to the 3D domain, intending to incrementally recognize new object categories from scarce labeled point cloud samples while effectively retaining knowledge of previously learned classes (Tao et al., 2020; Zhang et al., 2021; Yang et al., 2022a; Liu et al., 2024; Tian et al., 2024; Wu et al., 2025). While early studies in 2D vision explored strategies such as backbone fine-tuning (Ahmad et al., 2022; Liu et al., 2022; Kim et al., 2023; Liu et al., 2023; Lin et al., 2024), meta-learning-based prototype optimization (Chi et al., 2022; Hersche et al., 2022; Zhou et al., 2022b), and dynamic architecture adaptation (Zhang et al., 2021; Yang et al., 2022a; Zhao et al., 2023), recent advances have shifted toward parameter-efficient fine-tuning and multi-modal generalization. These include freezing pre-trained backbones with lightweight updates (Liu et al., 2024; Wang et al., 2024) and exploiting textual guidance for better generalization under data scarcity (Park et al., 2024; Sun et al.,

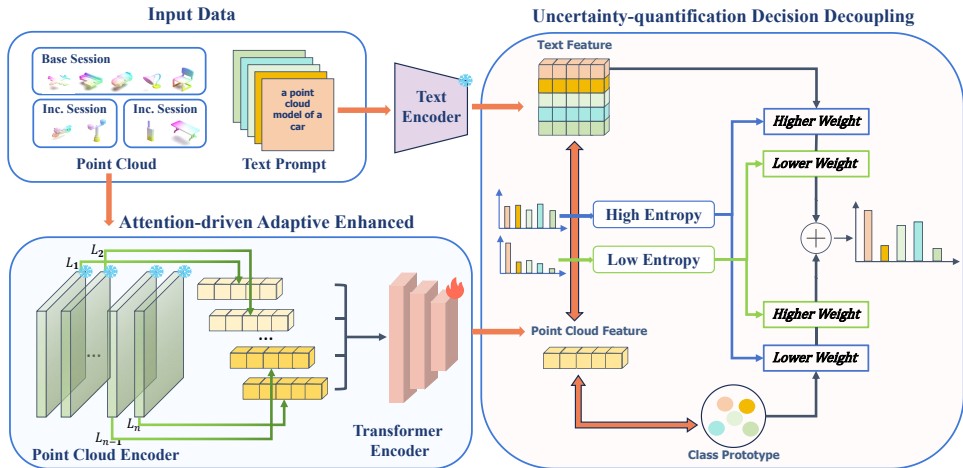

Figure 2: Overall architecture of Point-UQ. In the base-class training phase, the Attention-driven Adaptive Enhancement module integrates multi-scale features into calibrated representations via learnable multi-head self-attention, fusing shallow geometric details with deep semantic features. It simultaneously generates prediction entropy as a reliable metric of per-sample epistemic uncertainty, preserving semantic information while providing critical cues for subsequent decisions. During incremental learning, the Uncertainty-quantification Decision Decoupling module builds on AAE's outputs: using the precomputed entropy to quantify uncertainty, it dynamically balances weights between semantic classification and geometric prototype matching. For high-entropy ambiguous samples, it prioritizes geometric prototype matching to leverage base-class geometric priors and compensate for scarce novel samples. For low-entropy clear samples, it emphasizes semantic classifiers to reuse stable base-class decision boundaries. Point-UQ synergy enables robust knowledge retention and accurate novel-class recognition without retraining.

2024a). When applied to point cloud analysis, FSCIL introduces two distinct challenges: cross-domain distribution shifts and geometric detail degradation (Chowdhury et al., 2022). To address the former, Microshape (Chowdhury et al., 2022) proposes domain-invariant geometric descriptors that effectively bridge synthetic-to-real domain gaps. Building upon this idea, Cross-Domain (Tan & Xiang, 2024) further improves generalization by introducing dual-branch architectures that separately model distributions of old and new classes, enabling clearer decision boundaries during incremental updates. In addition to domain adaptation, recent works have explored visual-language pre-training to enhance semantic robustness in 3D continual learning. Methods such as C3PR (Cheraghian et al., 2025) and FoundationModel (Ahmadi et al., 2024), inspired by CLIP, transfer high-level semantic priors from 2D vision-language data to 3D domains, opening up new possibilities for knowledge transfer. These approaches provide strong semantic grounding, which complements geometric feature learning and supports better adaptation to novel categories with limited supervision.

## 3 METHOD

### 3.1 PROBLEM FORMULATION

We formulate the 3D FSCIL problem as a sequential learning process where a model progressively acquires knowledge of disjoint class sets with highly imbalanced sample sizes. The class evolution proceeds through $T$ stages, denoted as $\{S_1, S_2, \ldots, S_T\}$. The first stage $S_1$ contains base classes $\mathcal{C}_{\text{base}}$ with abundant training data. For subsequent stages $t \geq 2$, each $S_t$ introduces a set of novel classes $\mathcal{C}_{\text{novel}}^t$, where $\mathcal{C}_{\text{novel}}^t \cap \mathcal{C}_{\text{novel}}^{t'} = \emptyset$ for $t \neq t'$ and $\mathcal{C}_{\text{base}} \cap \mathcal{C}_{\text{novel}}^t = \emptyset$ for all $t$. Importantly, after training on $S_1$, base class data becomes inaccessible, mimicking real-world data privacy and storage constraints. Each class $c \in \mathcal{C}_{\text{base}} \cup \mathcal{C}_{\text{novel}}^t$ is associated with a set of 3D point clouds $X_c = \{x_{c1}, x_{c2}, \ldots, x_{cN_c}\}$, where $x_{ci} = \{p_{cij} \in \mathbb{R}^3\}_{j=1}^L$ represents a point cloud with $L$ Euclidean points. For novel classes in $S_t$, the sample size is severely limited, imposing the few-shot learning constraint.

## 3.2 POINT-UQ: POINT CLOUD UNCERTAINTY-QUANTIFICATION PARADIGM FOR FSCIL

The proposed Point-UQ framework reframes 3D FSCIL by shifting focus from feature tuning to decision optimization, operationalizing uncertainty quantification as the core of its unified pipeline (Fig. 2). Specifically, after a backbone network extracts hierarchical geometric features from 3D point clouds, AAE first fuses these multi-scale representations into calibrated features while generating prediction entropy as a reliable uncertainty metric. Building directly on these outputs, UDD then dynamically arbitrates between semantic classification and geometric prototype matching based on the entropy signals, enabling robust incremental learning without retraining. Detailed theoretical analysis supporting this framework is provided in Appendix J.

### 3.2.1 ATTENTION-DRIVEN ADAPTIVE ENHANCED MODULE

In 3D FSCIL, balancing base-class knowledge retention and novel-class adaptation requires representations that preserve both fine-grained geometric details and high-level semantic consistency (Tian et al., 2024; Zhao et al., 2024). Existing methods either underutilize shallow local features critical for distinguishing similar categories or lose detailed geometric cues during deep semantic aggregation, while fixed-weight fusion fails to adapt to dynamic discriminative demands (Tang et al., 2024). To address this, the AAE module employs learnable multi-head self-attention to dynamically fuse hierarchical features: it emphasizes task-relevant patterns across scales while suppressing noise. Critically, beyond producing calibrated representations, AAE leverages prediction entropy as a reliable estimate of per-sample epistemic uncertainty. This dual output—semantically preserved features paired with uncertainty cues—establishes the foundation for UDD's adaptive decision-making in subsequent incremental stages.

Let the backbone network produce hierarchical features $\mathcal{F} = \{F_l\}_{l=1}^{L}$, where shallow layers $F_1, F_2$ capture local geometric attributes and deep layers $F_{L-1}, F_L$ encode global shape semantics. These are stacked into a tensor:

$$\mathbf{F}_{\text{stack}} = \text{Stack}(F_1, F_2, \ldots, F_L) \in \mathbb{R}^{B \times L \times D}, \tag{1}$$

where $B$ is the batch size and $D$ is the feature dimension. The primary point cloud feature $\mathbf{F}_{\text{pc}} \in \mathbb{R}^{B \times D}$ is expanded to a query vector $\mathbf{Q} = \mathbf{F}_{\text{pc}} \in \mathbb{R}^{B \times 1 \times D}$, which is concatenated with $\mathbf{F}_{\text{stack}}$ to form:

$$\mathbf{F}_{\text{joint}} = \text{Concat}(\mathbf{Q}, \mathbf{F}_{\text{stack}}) \in \mathbb{R}^{B \times (L+1) \times D}, \tag{2}$$

Using multi-head self-attention, AAE models cross-level dependencies. For the $h$-th attention head, query, key, and value matrices are generated via linear projections:

$$\mathbf{Q}_h = \mathbf{F}_{\text{joint}} \mathbf{W}_h^Q, \quad \mathbf{K}_h = \mathbf{F}_{\text{joint}} \mathbf{W}_h^K, \quad \mathbf{V}_h = \mathbf{F}_{\text{joint}} \mathbf{W}_h^V, \tag{3}$$

where $\mathbf{W}_h^Q \in \mathbb{R}^{D \times d_k}$, $\mathbf{W}_h^K \in \mathbb{R}^{D \times d_k}$, and $\mathbf{W}_h^V \in \mathbb{R}^{D \times d_v}$ are learnable projection matrices ($d_k$ is the key/query dimension, $d_v$ is the value dimension). Attention weights are computed as:

$$A_h = \text{Softmax}\left(\frac{\mathbf{Q}_h \mathbf{K}_h^\top}{\sqrt{d_k}}\right) \in \mathbb{R}^{B \times (L+1) \times (L+1)}, \tag{4}$$

Concatenating head outputs and projecting through $\mathbf{W}_{\text{out}} \in \mathbb{R}^{H \cdot d_v \times D}$ yields the fused feature:

$$\mathbf{F}_{\text{fused}} = \text{Concat}(\text{Head}_1, \ldots, \text{Head}_H) \mathbf{W}_{\text{out}}, \quad \text{Head}_h = A_h \mathbf{V}_h, \tag{5}$$

To retain original local geometry, a residual connection combines the primary feature with the fused query-position feature:

$$\mathbf{F}_{\text{final}} = \mathbf{F}_{\text{pc}} + \mathbf{W}_{\text{res}} \cdot \mathbf{F}_{\text{fused}}^{(0)}, \tag{6}$$

where $\mathbf{F}_{\text{fused}}^{(0)}$ is the query-derived feature (first element of $\mathbf{F}_{\text{fused}}$) and $\mathbf{W}_{\text{res}} \in \mathbb{R}^{D \times D}$ is a learnable weight matrix. This design enhances fine-grained discriminability by integrating dynamic multi-scale interactions with local geometric preservation.

### 3.2.2 Uncertainty-quantification decision decoupling module

In 3D FSCIL, decision conflicts between base and novel classes represent a core performance bottleneck (Tian et al., 2024). Base classes, benefiting from abundant training samples, typically exhibit stable semantic feature distributions and confident classifier outputs. However, directly applying these classifiers to novel-class prediction—especially when fine-tuning is involved—can lead to catastrophic forgetting of base-class knowledge (Ahmadi et al., 2024). Conversely, novel classes—characterized by scarce samples—suffer from under-trained semantic classifier weights, whereas geometric prototype matching based on class centers provides a more robust discriminative basis through spatial structure analysis. UDD addresses this by decoupling classification strategies via prediction entropy, combining a semantic branch for high-confidence base class predictions and a geometric branch for novel class discrimination, with adaptive fusion guided by uncertainty.

The semantic classifier branch leverages pre-trained base classifiers to compute logits for base classes:

$$s_{\text{sem}} = f \cdot W_{\text{base}}^{\top} \in \mathbb{R}^{C_{\text{base}}}, \tag{7}$$

where $f \in \mathbb{R}^D$ is the AAE-enhanced feature and $W_{\text{base}} \in \mathbb{R}^{C_{\text{base}} \times D}$ are base classifier weights.

For the geometric prototype branch, robust prototypes are constructed to mitigate outlier effects and enhance semantic relevance. First, $K$-means clustering identifies the class centroid $\mu_c = \text{KMeans}(\mathcal{F}_c)$, and the $m$ closest samples (by Euclidean distance $d_i = \|\mathbf{f}_i - \mu_c\|_2$) are selected as core features $\mathcal{F}_c^{\text{sel}}$. Using class textual features $\mathbf{p}_c$, semantic similarity $s_j = \mathbf{f}_{i_j}^{\top} \mathbf{p}_c$ is computed and normalized via stabilized softmax to yield weights:

$$w_j = \frac{\exp(s_j - \max(s))}{\sum_{j=1}^{m} \exp(s_j - \max(s))}, \quad s = \{s_1, \ldots, s_m\}, \tag{8}$$

The final class prototype is obtained as a weighted average of the selected core features:

$$\mathbf{c}_c = \frac{\sum_{j=1}^{m} w_j \mathbf{f}_{i_j}}{\sum_{j=1}^{m} w_j}, \tag{9}$$

During inference, cosine similarity scores between the test feature $f \in \mathbb{R}^D$ and each novel class prototype $\mathbf{c}_k$ are computed as:

$$s_{\text{geo}}(k) = \frac{f \cdot \mathbf{c}_k}{\|f\| \cdot \|\mathbf{c}_k\|}, \quad \forall k \in \{1, \ldots, C_{\text{novel}}\}, \tag{10}$$

To quantify uncertainty in the semantic branch output, prediction entropy is used:

$$H(p) = -\sum_c p_c \log p_c, \quad p = \text{Softmax}(s_{\text{sem}}), \tag{11}$$

where $s_{\text{sem}} \in \mathbb{R}^{C_{\text{base}}}$ denotes the logits output from the semantic classifier. A tunable scaling factor $\lambda$ adjusts the fusion weight via the sigmoid function:

$$\alpha = \sigma(\lambda \cdot H(p)) \in [0, 1], \tag{12}$$

Finally, the classification score is adaptively fused from both branches:

$$s_{\text{final}} = \alpha \cdot s_{\text{geo}} + (1 - \alpha) \cdot s_{\text{sem}}, \tag{13}$$

This mechanism prioritizes semantic confidence for clear samples while relying more on geometric matching under high uncertainty, effectively reducing catastrophic forgetting and misclassification.

### 3.3 Training pipeline

The model is trained solely on base classes with frozen point cloud and text encoders, updating only AAE and UDD prototype construction parameters, with training guided by two loss functions:

The cross-entropy loss ensures classification accuracy:

$$\mathcal{L}_{\text{ce}} = -\frac{1}{N} \sum_{i=1}^{N} \sum_{c=1}^{C} y_{ic} \log(p_{ic}), \tag{14}$$

where $y_{ic}$ is the one-hot label and $p_{ic}$ is the predicted probability. The cosine similarity loss aligns features with their class prototypes:

$$\mathcal{L}_{\text{cos}} = 1 - \frac{1}{N} \sum_{i=1}^{N} \frac{\mathbf{f}_i^\top \mathbf{c}_{y_i}}{\|\mathbf{f}_i\| \cdot \|\mathbf{c}_{y_i}\|}, \tag{15}$$

where $\mathbf{c}_{y_i}$ is the prototype of the ground-truth class. The total loss balances these components with a tunable weight $\beta$:

$$\mathcal{L}_{\text{total}} = \beta \cdot \mathcal{L}_{\text{ce}} + \mathcal{L}_{\text{cos}}. \tag{16}$$

This setup ensures the model learns discriminative geometric-semantic representations while maintaining base class knowledge through prototype regularization.

# 4 EXPERIMENT

Table 1: Average Accuracy on Intra-dataset Evaluations. Each column (e.g., "20-40") shows the cumulative number of classes learned, with $\mathbf{\Delta}_\downarrow$ quantifying the relative drop in overall average accuracy by the final session.

| Method | ModelNet | | | | | | CO3D | | | | | | | ShapeNet | | | | | | | |
|---|---|---|---|---|---|---|---|---|---|---|---|---|---|---|---|---|---|---|---|---|---|
| | 20 | 25 | 30 | 35 | 40 | $\mathbf{\Delta}_\downarrow$ | 25 | 30 | 35 | 40 | 45 | 50 | $\mathbf{\Delta}_\downarrow$ | 25 | 30 | 35 | 40 | 45 | 50 | 55 | $\mathbf{\Delta}_\downarrow$ |
| FT | 89.8 | 9.7 | 4.3 | 3.3 | 3.0 | 96.7 | 76.7 | 11.2 | 3.6 | 3.2 | 1.8 | 0.8 | 99.0 | 87.0 | 25.7 | 6.8 | 1.3 | 0.9 | 0.6 | 0.4 | 99.5 |
| Joint | 89.8 | 88.2 | 87.0 | 83.5 | 80.5 | 10.4 | 76.7 | 69.4 | 64.8 | 62.7 | 60.7 | 59.8 | 22.0 | 87.0 | 85.2 | 84.3 | 83.0 | 82.5 | 82.2 | 81.3 | 6.6 |
| LwF | 89.8 | 36.0 | 9.1 | 3.6 | 3.1 | 96.0 | 76.7 | 14.7 | 4.7 | 3.5 | 2.3 | 1.0 | 98.7 | 87.0 | 60.8 | 33.5 | 15.9 | 3.8 | 3.1 | 1.8 | 97.9 |
| IL2M | 89.8 | 65.5 | 58.4 | 52.3 | 53.6 | 40.3 | 76.7 | 31.5 | 27.7 | 18.1 | 27.1 | 21.9 | 71.4 | 87.0 | 58.6 | 45.7 | 40.7 | 50.1 | 49.4 | 49.3 | 43.3 |
| ScaIL | 89.8 | 66.8 | 64.5 | 58.7 | 56.5 | 37.1 | 76.7 | 39.5 | 34.1 | 24.1 | 30.1 | 27.5 | 64.1 | 87.0 | 56.6 | 51.8 | 44.3 | 50.3 | 46.3 | 45.4 | 47.8 |
| EEIL | 89.8 | 75.4 | 67.2 | 60.1 | 55.6 | 38.1 | 76.7 | 61.4 | 52.4 | 42.8 | 39.5 | 32.8 | 57.2 | 87.0 | 77.7 | 73.2 | 69.3 | 66.4 | 65.9 | 65.8 | 22.4 |
| FACT | 90.4 | 81.3 | 77.1 | 73.5 | 65.0 | 28.1 | 77.9 | 67.1 | 59.7 | 54.8 | 50.2 | 46.7 | 40.0 | 87.5 | 75.3 | 71.4 | 69.9 | 67.5 | 65.7 | 62.5 | 28.6 |
| Sem-aware | 91.3 | 82.2 | 74.3 | 70.0 | 64.7 | 29.1 | 78.6 | 66.9 | 59.2 | 53.6 | 49.1 | 42.9 | 44.1 | 87.2 | 74.9 | 68.1 | 69.0 | 68.1 | 66.9 | 63.8 | 26.8 |
| Microshape | 93.6 | 83.1 | 78.2 | 75.8 | 67.1 | 28.3 | 78.5 | 67.3 | 60.1 | 56.1 | 51.4 | 47.2 | 39.9 | 87.6 | 83.2 | 81.5 | 79.0 | 76.8 | 73.5 | 72.6 | 17.1 |
| C3PR | 91.6 | 82.3 | 75.8 | 72.2 | 70.9 | 22.5 | 81.5 | 69.4 | 66.5 | 63.0 | 54.2 | 53.8 | 34.0 | 88.0 | 81.6 | 77.8 | 76.7 | 76.9 | 76.2 | 74.7 | 15.1 |
| **Point-UQ(ours)** | **97.1** | **91.0** | **86.8** | **83.0** | **79.0** | **18.6** | **82.1** | **73.6** | **70.0** | **68.4** | **67.4** | **66.5** | **19.0** | **93.3** | **91.2** | **90.0** | **89.3** | **88.8** | **88.4** | **86.5** | **7.0** |

Table 2: Average Accuracy on Cross-dataset Evaluations. Each column (e.g., "39-89") shows the cumulative number of classes learned, with $\mathbf{\Delta}_\downarrow$ quantifying the relative drop in overall average accuracy by the final session.

| Method | ShapeNet → CO3D | | | | | | | | | | | | ModelNet → ScanObjectNN | | | | | ShapeNet → ScanObjectNN | | | | |
|---|---|---|---|---|---|---|---|---|---|---|---|---|---|---|---|---|---|---|---|---|---|---|
| | 39 | 44 | 49 | 54 | 59 | 64 | 69 | 74 | 79 | 84 | 89 | $\mathbf{\Delta}_\downarrow$ | 26 | 30 | 34 | 37 | $\mathbf{\Delta}_\downarrow$ | 44 | 49 | 54 | 59 | $\mathbf{\Delta}_\downarrow$ |
| FT | 81.0 | 20.2 | 2.3 | 1.7 | 0.8 | 1.0 | 1.0 | 1.3 | 0.9 | 0.5 | 1.6 | 98.0 | 88.4 | 6.4 | 6.0 | 1.9 | 97.9 | 81.4 | 38.7 | 4.0 | 0.9 | 98.9 |
| Joint | 81.0 | 79.5 | 78.3 | 75.2 | 75.1 | 74.8 | 72.3 | 71.3 | 70.0 | 68.8 | 67.3 | 16.9 | 88.4 | 79.7 | 74.0 | 71.2 | 19.5 | 81.4 | 82.5 | 79.8 | 78.7 | 3.3 |
| LwF | 81.0 | 57.4 | 19.3 | 2.3 | 1.0 | 0.9 | 0.8 | 1.3 | 1.1 | 0.8 | 1.9 | 97.7 | 88.4 | 35.8 | 5.8 | 2.5 | 97.2 | 81.4 | 47.9 | 14.0 | 5.9 | 92.8 |
| IL2M | 81.0 | 45.6 | 36.8 | 35.1 | 31.8 | 33.3 | 34.0 | 31.5 | 30.6 | 32.3 | 30.0 | 63.0 | 88.4 | 58.2 | 52.9 | 52.0 | 41.2 | 81.4 | 53.2 | 43.9 | 45.8 | 43.7 |
| ScaIL | 81.0 | 50.1 | 45.7 | 39.1 | 39.0 | 37.9 | 38.0 | 36.0 | 33.7 | 33.0 | 35.2 | 56.5 | 88.4 | 56.5 | 55.9 | 52.9 | 40.2 | 81.4 | 49.0 | 46.7 | 40.0 | 50.9 |
| EEIL | 81.0 | 75.2 | 69.3 | 63.2 | 60.5 | 57.9 | 53.0 | 51.9 | 51.3 | 47.8 | 47.6 | 41.2 | 88.4 | 70.2 | 61.0 | 56.8 | 35.7 | 81.4 | 74.5 | 69.8 | 63.4 | 22.1 |
| FACT | 81.4 | 76.0 | 70.3 | 68.1 | 65.8 | 63.5 | 63.0 | 60.1 | 58.2 | 57.5 | 55.9 | 31.3 | 89.1 | 72.5 | 68.3 | 63.5 | 28.7 | 82.3 | 74.6 | 69.9 | 66.8 | 18.8 |
| Sem-aware | 80.6 | 69.5 | 66.5 | 62.9 | 63.2 | 63.0 | 61.2 | 58.3 | 58.1 | 57.2 | 55.2 | 31.6 | 88.5 | 73.9 | 67.7 | 64.2 | 27.5 | 81.3 | 70.6 | 65.2 | 62.9 | 22.6 |
| Microshape | 82.6 | 77.9 | 73.9 | 72.7 | 67.7 | 66.2 | 65.4 | 63.4 | 60.6 | 58.1 | 57.1 | 30.9 | 89.3 | 73.2 | 68.4 | 65.1 | 27.1 | 82.5 | 74.8 | 71.2 | 67.1 | 18.7 |
| C3PR | 83.6 | 80.0 | 77.8 | 75.4 | 72.8 | 72.3 | 70.3 | 67.9 | 64.9 | 64.1 | 63.2 | 24.4 | 88.3 | 75.7 | 70.6 | 67.8 | 23.2 | 84.5 | 77.8 | 75.5 | 71.9 | 14.9 |
| FoundationModel | 87.3 | 86.2 | 84.4 | 82.2 | 80.7 | 79.6 | 78.2 | 76.8 | 76.1 | 74.5 | 72.6 | 16.8 | 87.7 | 84.7 | 81.5 | 79.2 | 9.6 | 90.8 | 86.5 | 86.4 | 85.6 | 5.7 |
| **Point-UQ(ours)** | **91.4** | **89.5** | **88.3** | **86.5** | **84.7** | **83.4** | **82.9** | **81.8** | **81.1** | **80.4** | **80.3** | **12.1** | **93.1** | **90.7** | **87.2** | **86.6** | **7.0** | **90.9** | **88.2** | **86.8** | **86.0** | **5.4** |

## 4.1 EXPERIMENTAL SETUP

**Dataset partitioning and implementation details.** To evaluate the proposed 3D FSCIL method, we conduct experiments on four widely-used datasets, following the same setup as Microshape (Chowdhury et al., 2022), including both intra-dataset and cross-dataset settings. For partition details and comprehensive experimental implementation information, refer to Appendix A and Appendix B.

**Evaluation metrics.** To comprehensively evaluate the model's ability to retain knowledge of base classes while learning new ones, we adopt three key metrics: Average Accuracy, Relative Accuracy

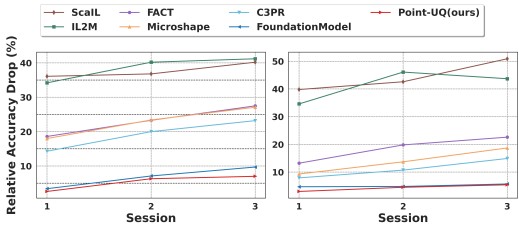 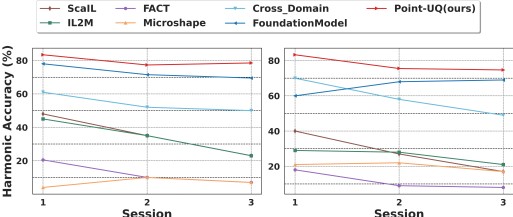

Figure 3: Experimental results of relative accuracy drop on ModelNet → ScanObjectNN (Left) and ShapeNet → ScanObjectNN (Right).

Figure 4: Experimental results of harmonic accuracy on ModelNet → ScanObjectNN (Left) and ShapeNet → ScanObjectNN (Right).

Drop Rate ($\Delta$), and Harmonic Accuracy (Peng et al., 2022). These metrics jointly reflect the trade-off between minimizing catastrophic forgetting and maximizing novel-class adaptation. For detailed formulations of these evaluation metrics, please refer to Appendix C.

**Compared methods.** To evaluate our approach, we compare it against three categories of baseline methods. Conventional Transfer Learning Methods: Fine-tuning, which updates the model using only new class data, and joint training, where the model accesses all previous and new class data to mitigate forgetting. Adapted 2D FSCIL Methods: We adapt successful 2D FSCIL approaches for 3D point cloud data by replacing their CNNs with PointNet, assessing their applicability in 3D contexts (Belouadah & Popescu, 2019; 2020; Castro et al., 2018; Li & Hoiem, 2017; Zhou et al., 2022a; Cheraghian et al., 2021). Specialized 3D FSCIL Methods: Techniques specifically designed for 3D FSCIL tasks, including Microshape (Chowdhury et al., 2022), Cross-Domain (Tan & Xiang, 2024), C3PR (Cheraghian et al., 2025), and FoundationModel (Ahmadi et al., 2024), addressing unique challenges posed by point cloud data.

## 4.2 EXPERIMENTAL PERFORMANCE ANALYSIS

In Tab. 1 and Tab. 2, we present the comparison results for within-dataset and cross-dataset settings, respectively. [1]

**Intra-dataset and cross-dataset performance evaluation.** In intra-dataset settings, 2D-adapted methods fail to capture point cloud geometric structures due to reliance on CNN-style global features. Even dedicated 3D baselines like C3PR (Cheraghian et al., 2025) and FoundationModel (Ahmadi et al., 2024) lack fine-grained discrimination—they depend solely on deep semantic features and overlook geometry, causing notable accuracy drops on sparse samples. In contrast, Point-UQ achieves a consistent 4% accuracy gain at each incremental stage, rooted in our core innovation: uncertainty-guided decision optimization paired with explicit 3D geometry modeling. The AAE module boosts shape discriminability for subtle differences via multi-scale feature integration, while the UDD module curbs misclassification of ambiguous samples.

In cross-dataset tasks, 2D-adapted methods suffer up to 20% novel-class accuracy collapse due to lacking 3D invariance. Dedicated 3D baselines also perform poorly under domain shifts, failing to adapt to variations in point cloud density and noise. Point-UQ, however, excels by design: its decision logic is inherently adaptive. The AAE module dynamically fuses multi-level geometric features using learnable attention weights, tailoring representations to the current task context. More importantly, the UDD mechanism exploits uncertainty as a switching signal—diverting high-entropy samples to geometry-based matching, which exhibits superior cross-domain stability. This shift from fixed to uncertainty-aware decision-making allows Point-UQ to achieve state-of-the-art performance, surpassing the second-best method by 9%, while maintaining strong base-class recognition.

**Forgetting and adaptation efficiency.** To more accurately reflect the model's forgetting behavior during incremental learning, we further calculated the accuracy drop relative to the base classes at each incremental stage (Fig. 3). Our method records the smallest average drop, nearly two times lower than baselines such as Microshape (Chowdhury et al., 2022). Furthermore, using the *harmonic mean*

---

[1]The results of the compared methods listed in the tables are sourced from (Ahmadi et al., 2024) and (Cheraghian et al., 2025), and all experiments were conducted under consistent configurations. Notably, the FoundationModel was omitted from Tab. 1 as its original paper did not report intra-dataset performance, and we encountered significant discrepancies when attempting to reproduce its results.

*metric*—which balances performance between base and novel classes—our method outperforms existing approaches by 9% on average across cross-dataset benchmarks (Fig. 4). These results validate that integrating geometric structure modeling with uncertainty-aware decision-making addresses critical limitations in prior work, setting new standards for 3D few-shot class incremental learning. Meanwhile, we also conducted an analysis of the computational complexity of the method, comparing it from two aspects: time efficiency and cache efficiency. Detailed experimental analysis can be found in Appendix D.

## 4.3 ABLATION STUDY

To rigorously assess the contribution of each component in our framework, we carried out ablation experiments on the cross-dataset setting from ShapeNet to CO3D. The performance was evaluated using both average accuracy and harmonic accuracy.

Table 3: Effectiveness of the components in Point-UQ.

| AAE | UDD | Average Accuracy (%) | | | | | | | | | | | Harmonic Acc Avg (%) |
|---|---|---|---|---|---|---|---|---|---|---|---|---|---|
| | | 0 | 1 | 2 | 3 | 4 | 5 | 6 | 7 | 8 | 9 | 10 | Avg |
| ✗ | ✗ | 87.9 | 86.3 | 85.2 | 82.6 | 80.3 | 79.3 | 77.9 | 76.6 | 76.1 | 74.6 | 73.6 | 54.5 |
| ✓ | ✗ | 89.0 | 88.5 | 86.8 | 85.1 | 84.1 | 81.4 | 78.2 | 79.5 | 77.8 | 76.1 | 75.1 | 55.7 |
| ✗ | ✓ | 89.8 | 88.5 | 87.1 | 85.2 | 83.6 | 81.9 | 81.2 | 80.3 | 79.7 | 79.2 | 78.9 | 60.0 |
| ✓ | ✓ | **91.4** | **89.5** | **88.3** | **86.5** | **84.7** | **83.4** | **82.9** | **81.8** | **81.1** | **80.4** | **80.3** | **64.8** |

**Evaluating component effectiveness in Point-UQ.** To evaluate the effectiveness of each component in Point-UQ, we conducted a series of experiments. First, we evaluate the effectiveness of our two proposed modules: Attention-driven Adaptive Enhanced Module and Uncertainty-quantification Decision Decoupling Module, with results summarized in Tab. 3. Ablation studies are conducted by replacing AAE with a standard alignment mechanism and substituting UDD with a dual-feature caching strategy (Ahmadi et al., 2024). Removing AAE leads to a notable performance drop in the base-class training phase, highlighting its importance in capturing and fusing multi-scale geometric features. The UDD module dynamically balances semantic and geometric prototype similarities based on prediction entropy, enabling uncertainty-aware soft fusion. This allows the model to adaptively leverage reliable knowledge sources during novel class recognition.

In addition to the ablation studies on our main components, we further evaluate the effectiveness of the proposed multi-scale self-attention feature fusion mechanism by comparing it with several representative feature fusion strategies, and validate the design of our semantic-weighted class prototype construction approach through dedicated analysis as shown in Fig. 5. The experimental results of the proposed multi-scale self-attention feature fusion mechanism are shown in Tab. 4. Specifically, "Deep-Semantic-only" refers to using only the high-level semantic features from the final layer for prediction. "LayerWise-To-Last" introduces shallow features and fuses each layer with the last one individually, generating separate predictions which are then aggregated for the final output (Tang et al., 2024). "Symmetric-Cross-Fusion" also incorporates shallow features but adopts a symmetric layer-wise fusion strategy—pairing the first with the last, the second with the second-to-last, and so on. In contrast, our method, Point-UQ, employs a multi-scale self-attention mechanism to adaptively integrate shallow geometric details with deep semantic features, enabling more effective cross-level information interaction. The results demonstrate that, compared to fixed-rule-based fusion methods, our self-attention fusion mechanism can dynamically adjust the contribution weights of different feature levels according to task demands. This not only improves the model's discriminative capability but also enhances robustness to class variations. These findings further confirm the superiority and necessity of Point-UQ in addressing complex feature distribution challenges in incremental learning scenarios.

Table 4: Effectiveness of the proposed multi-scale self-attention feature fusion mechanism.

| Method | Average Accuracy (%) | | | | | | | | | | | Harmonic Acc Avg (%) |
|---|---|---|---|---|---|---|---|---|---|---|---|---|---|
| | 0 | 1 | 2 | 3 | 4 | 5 | 6 | 7 | 8 | 9 | 10 | Avg |
| Deep-Semantic-only | 89.8 | 88.5 | 87.1 | 85.2 | 83.6 | 81.9 | 81.2 | 80.3 | 79.7 | 79.2 | 78.9 | 60.0 |
| LayerWise-To-Last | 90.5 | 89.1 | 87.5 | 85.7 | 83.9 | 82.5 | 82.1 | 81.1 | 80.4 | 80.0 | 79.8 | 61.0 |
| Symmetric-Cross-Fusion | 90.1 | 88.9 | 87.3 | 85.0 | 83.6 | 82.0 | 81.6 | 80.5 | 80.0 | 79.4 | 79.2 | 59.4 |
| Point-UQ(ours) | **91.4** | **89.5** | **88.3** | **86.5** | **84.7** | **83.4** | **82.9** | **81.8** | **81.1** | **80.4** | **80.3** | **64.8** |

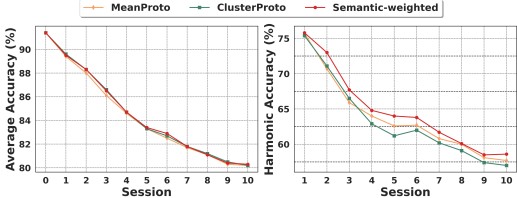 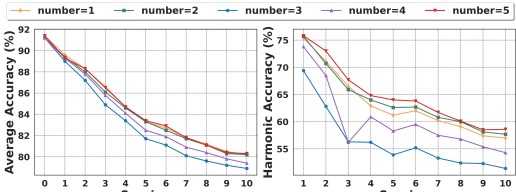

Figure 5: Experimental results of the Proposed Semantic-weighted Class Prototype Construction Approach.

Figure 6: Impact of the Number of Samples Used for Constructing Semantic-weighted Class Prototypes on Final Performance.

We further validate the effectiveness of our semantic-weighted class prototype construction method, as illustrated in Fig. 5. In uncertainty quantification, well-constructed class prototypes are essential, as our method dynamically assigns higher weight to geometric similarity when prediction entropy is high—thereby leveraging structural relationships among prototypes to enhance decision reliability. To evaluate the advantage of our strategy, we compare it against two widely used baseline methods: **MeanProto**: Computes the class prototype as the mean of all sample features within that class. **ClusterProto**: Selects representative central points via clustering to serve as class prototypes. Experimental results demonstrate that incorporating semantic information into feature weighting yields significantly more discriminative and robust prototypes. Our method achieves superior performance in both novel class recognition and uncertainty estimation, further validating its effectiveness in modeling dynamic class representations during incremental learning.

**Hyperparameter experiments.** In Point-UQ, three key hyperparameters are involved: the number of samples used to construct semantic-weighted class prototypes $m$, the weight coefficient that balances the cross-entropy loss and cosine loss $\beta$, and the activation function coefficient that maps entropy values to weights $\lambda$. To evaluate their influence on overall performance, we conduct a comprehensive ablation study in this section. Fig. 6 illustrates the impact of the sample count on the final performance, while the remaining experimental results and detailed analysis can be found in Appendix F. Besides, we provide extended ablation studies about activation function and alternative uncertainty measures in Appendix G.

## 5 CONCLUSION

In this paper, we propose Point-UQ, a novel uncertainty-quantification framework for 3D FSCIL. Departing from fine-tuning-focused paradigms, it integrates two synergistic modules: AAE, which fuses multi-scale features and generates epistemic uncertainty via prediction entropy; and UDD, which dynamically arbitrates between semantic classification and geometric matching. This design enables Point-UQ to balance base-class retention and novel-class adaptation without retraining, mitigating catastrophic forgetting and overfitting. Extensive experiments across multiple 3D benchmarks confirm that our approach consistently outperforms state-of-the-art methods in terms of accuracy, forgetting rate, and harmonic mean, establishing new performance standards for 3D FSCIL.

## 6 SUPPLEMENTARY STATEMENTS

### 6.1 ETHICS STATEMENT

This research strictly adheres to the ICLR Code of Ethics with no ethics-related risks: it uses public 3D point cloud benchmark datasets (ModelNet, ShapeNet, ScanObjectNN, CO3D) and focuses on algorithmic innovation for 3D few-shot class-incremental learning, without involving scenarios endangering public safety, infringing privacy, or producing discrimination.

### 6.2 REPRODUCIBILITY STATEMENT

To ensure reproducibility, dataset partitioning, experimental configurations, method details, and evaluation metrics are thoroughly described in the experimental section and corresponding appendices. Experimental results of comparative methods are sourced from public literature, and our experiments strictly follow the same configurations as baseline methods for fair comparison.

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

## A APPENDIX FOR DATASET PARTITIONING

To better evaluate the practical applicability of Point-UQ under real-world constraints, we design a series of cross-dataset incremental learning experiments that simulate scenarios where novel categories come from limited and noisy real-scanned data. These setups emphasize generalization across domains and robustness to data scarcity—key challenges in few-shot class incremental learning (FSCIL) for 3D point clouds.

In the most realistic setting, we transfer knowledge from clean synthetic data to real-world scans. Specifically, we first train on base classes from ModelNet or ShapeNet and incrementally introduce new classes from ScanObjectNN—a dataset derived from real-world scans with inherent noise and incompleteness. Following Chowdhury et al. (2022), we define four incremental tasks when transitioning from ModelNet to ScanObjectNN. For the ShapeNet to ScanObjectNN setup, we use 44 base classes from ShapeNet and introduce 15 real-scanned classes from ScanObjectNN across four stages. The most challenging scenario is established between ShapeNet and CO3D, where we define eleven incremental tasks. In this setting, 44 base classes are taken from ShapeNet, while 50 fine-grained object classes from CO3D are introduced incrementally. This large-scale cross-domain transition poses significant difficulties due to the domain gap and complex intra-class variations present in CO3D. As a complement to these cross-dataset evaluations, we also conduct within-dataset experiments to establish baseline performance. On ModelNet, we split the dataset into 20 base classes and 20 incremental classes, distributed over four stages. For ShapeNet and CO3D, we adopt larger base sets: 25 base classes for each, with the remaining 30 (ShapeNet) or 25 (CO3D) classes divided into 7 or 6 incremental tasks, respectively.

These diverse experimental settings allow us to comprehensively assess our method's ability to generalize across both intra- and inter-dataset boundaries, particularly under data-limited and domain-shifted conditions.

## B APPENDIX FOR IMPLEMENTATION DETAILS

Our framework leverages the Uni3D model (Zhou et al., 2023) for both point cloud and text encoding, maintaining these components frozen throughout the training process. This design choice enables efficient few-shot learning without necessitating any parameter updates to the foundational models. Instead, we focus on optimizing the parameters of the AAE module, which consists of a 2-layer Transformer encoder with each layer employing an 8-head multi-head attention mechanism. To balance computational efficiency with performance, our setup mirrors that of the FoundationModel (Ahmadi et al., 2024), ensuring optimal resource utilization. Specifically, we adopt the "EVA02-E-14+" CLIP model alongside the "eva02-base_patch14_448" model as our point cloud encoders (Zhou et al., 2023). As input, we uniformly sample 1024 points from each 3D point cloud object using farthest point sampling. All experiments were conducted on a single NVIDIA A100 GPU. We utilized the AdamW optimizer with a weight decay of $1 \times 10^{-4}$ to manage overfitting. The training process spanned 8 epochs, employing a fixed learning rate of 0.00002 and a batch size of 32. In implementation, we fix the following hyperparameters across all experiments: the number of samples used to construct semantic-weighted class prototypes $m$ is set to 5; the coefficient balancing cross-entropy loss and cosine loss $\beta$ is set to 5; and the activation function coefficient that maps entropy values to weights $\lambda$ is set to 2.

## C APPENDIX FOR EVALUATION METRICS

To comprehensively evaluate the model's ability to retain knowledge of base classes while adapting to new ones, we adopt three key metrics: **harmonic accuracy** ($A_h$), **relative accuracy drop rate** ($\Delta$), and **average accuracy** (AA). These metrics jointly reflect the model's stability, adaptability, and overall performance during incremental learning.

- **Harmonic accuracy** ($A_h$): This metric balances performance on base and novel classes by computing the harmonic mean of their respective accuracies:

$$A_h = \frac{2 \times A_b \times A_n}{A_b + A_n}, \tag{C1}$$

where $A_b$ and $A_n$ denote the average accuracy on base and novel classes, respectively. A higher $A_h$ indicates a better trade-off between preserving old knowledge and learning new concepts (Peng et al., 2022).

- **Relative accuracy drop rate ($\Delta_\downarrow$)**: To quantify performance degradation caused by incremental updates, we compute the relative drop in total accuracy from the initial stage to the final one:

$$\Delta_\downarrow = \left| \frac{\text{acc}_T - \text{acc}_0}{\text{acc}_0} \right| \times 100. \tag{C2}$$

Here, $\text{acc}_T$ and $\text{acc}_0$ represent the total accuracy after the last and first incremental tasks, respectively. A smaller $\Delta_\downarrow$ suggests greater robustness against catastrophic forgetting.

- **Average accuracy (AA)**: As an overall measure of classification performance, we calculate the average accuracy across all seen classes at each incremental stage:

$$AA = \frac{\text{correctly predicted instances}}{\text{total instances in the test set}}. \tag{C3}$$

This provides a straightforward view of how well the model performs as new classes are introduced over time.

These three metrics together offer a comprehensive evaluation of model behavior—covering both class retention and adaptation capabilities, as well as overall classification accuracy.

## D    APPENDIX FOR COMPUTATIONAL COMPLEXITY

Our method's computational overhead from multi-head attention and clustering mainly occurs during base-class training, yet total training time remains shorter than FoundationModel (Ahmadi et al., 2024). In the incremental phase, the few-shot setup (5 samples per novel class) eliminates the need for clustering, and it is training-free—ideal for real-time scenarios.

Tab.D1 presents comparative experiments between Point-UQ and current SOTA on the cross-dataset setting from ShapeNet to CO3D, where FoundationModel (Ahmadi et al., 2024) is implemented strictly in accordance with its original paper. It is evident that our method achieves significant performance gains across all scenarios while using only one-fifth of the sample cache, with the average inference time per session extending by approximately 30 seconds.

Table D1: Comparison of Time and Memory Efficiency with SOTA

| Method | Time Efficiency | | Memory Efficiency |
|---|---|---|---|
| | Training Time (s) | Inference per Session (s) | Exemplars to Cache |
| FoundationModel | 2571 | **130** | 445 |
| Point-UQ | **1622** | 161 | **89** |

## E    APPENDIX FOR HYPERPARAMETER SENSITIVITY

Table E1: Impact of the entropy-to-weight mapping coefficient ($\lambda$) on experimental results.

| | Average Accuracy (%) | | | | | | | | | | | Harmonic Acc Avg (%) |
|---|---|---|---|---|---|---|---|---|---|---|---|---|
| $\lambda$ | 0 | 1 | 2 | 3 | 4 | 5 | 6 | 7 | 8 | 9 | 10 | Avg |
| 0.5 | 91.2 | 87.3 | 84.5 | 81.9 | 79.0 | 77.1 | 76.0 | 75.0 | 74.6 | 74.1 | 73.9 | 10.9 |
| 1.0 | 91.2 | 88.8 | 86.9 | 84.9 | 82.7 | 81 | 80.1 | 79.2 | 79.0 | 78.5 | 78.4 | 46.7 |
| 1.5 | 91.2 | **89.6** | 88.0 | 86.2 | 84.2 | 83.0 | 82.3 | 81.4 | 80.9 | 80.3 | 80.0 | 60.6 |
| 2.0 | **91.4** | 89.5 | **88.3** | **86.5** | **84.7** | **83.4** | **82.9** | **81.8** | **81.1** | **80.4** | **80.3** | 64.8 |
| 2.5 | 91.3 | 89.0 | 87.9 | 86.1 | 84.4 | 83.2 | 82.5 | 81.5 | 80.9 | 80.1 | 79.8 | 65.8 |
| 3.0 | 91.3 | 88.7 | 87.7 | 85.9 | 84.2 | 83.0 | 82.3 | 81.2 | 80.5 | 79.8 | 79.6 | **66.5** |

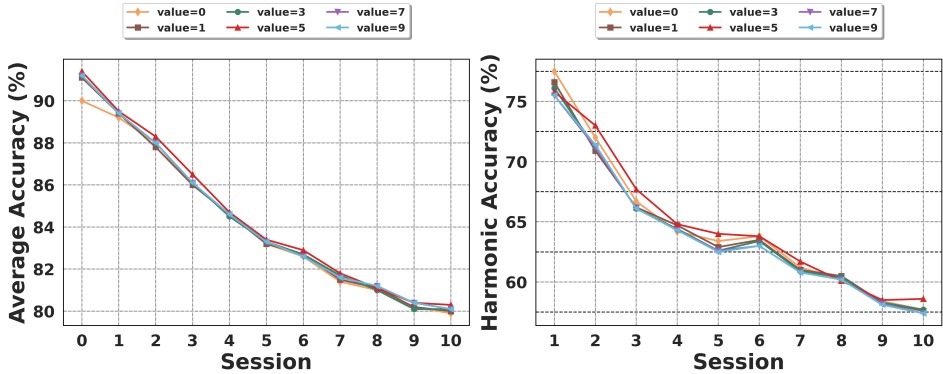

Figure E1: Impact of the loss balancing coefficient ($\beta$) on model performance.

In our method, three key hyperparameters are involved: the number of samples used to construct semantic-weighted class prototypes $m$, the weight coefficient that balances the cross-entropy loss and cosine loss $\beta$, and the activation function coefficient that maps entropy values to weights $\lambda$. To evaluate their influence on overall performance, we conduct a comprehensive ablation study in this section.

**Sample count** ($m$): As illustrated in Fig. 6, setting $m = 5$ yields the best performance across multiple evaluation metrics. With only five samples per novel class, excluding any through clustering ($m < 5$) risks discarding critical geometric patterns in sparse 3D data, which are essential for accurate prototype construction. This leads to unstable class representations and degrades generalization to new categories.

**Loss balancing coefficient** ($\beta$): During the base-class training stage, both cross-entropy loss and cosine loss are utilized to encourage discriminative feature learning. The trade-off between these two objectives is controlled by the balancing coefficient $\beta$. As shown in Fig. E1, we evaluate $\beta$ values ranging from 0 to 9. A value of $\beta = 5$ achieves the optimal performance because it allows for effective learning of both classification accuracy (via cross-entropy loss) and feature separation (via cosine loss). When $\beta$ is too small, the model overly emphasizes the cosine loss, which can hinder the alignment between point cloud features and textual embeddings—critical for cross-modal generalization. Conversely, when $\beta$ is too large, the cosine loss becomes underweighted, leading to less structured feature distributions and weakening the geometric prototype's supervisory role in shaping the embedding space.

**Entropy-to-weight mapping coefficient** ($\lambda$): The activation function coefficient $\lambda$ plays a crucial role in the UDD module, as it controls how uncertainty estimates are mapped to adaptive weights for the semantic decision branch and geometric prototype branch. Tab. E1 presents the results under different $\lambda$ settings, ranging from 0.5 to 3.0 with intervals of 0.5. A value of $\lambda = 2$ is found to yield the best performance. This suggests an appropriate sensitivity to uncertainty—neither too flat nor too steep in the weighting function. When $\lambda$ is too small, the mapping becomes overly sensitive to minor variations in entropy, causing instability in the final prediction. On the other hand, excessively large $\lambda$ values, while yielding high accuracy on novel categories, may cause forgetting of base-class knowledge, suppress low-uncertainty predictions, potentially discarding reliable decisions made by the model. With $\lambda = 2$, the model strikes a favorable balance between responsiveness to uncertainty and confidence in accurate predictions, which is essential for maintaining high performance without additional parameter updates during the incremental phase.

## F    EXTENDED EXPERIMENTAL ANALYSIS

### F.1    INTRA-DATASET BENCHMARKING

To ensure a comprehensive comparison, we evaluated the baseline method FoundationModel (Ahmadi et al., 2024) under an intra-dataset setting. As the original work did not report results for this setting

and the code was not available, we re-implemented it based on the methodological description and conducted extensive hyperparameter tuning to achieve its best attainable performance.

Tab. F1 shows that Point-UQ consistently and significantly outperforms the re-implemented FoundationModel across all datasets and incremental stages, even under this favorable configuration for the baseline.

Table F1: Comparative performance (%) under the intra-dataset setting.

| Dataset | Method | 0 | 1 | 2 | 3 | 4 | 5 | 6 | $\Delta_\downarrow$ |
|---------|--------|-----|-----|-----|-----|-----|-----|-----|------|
| ShapeNet | FoundationModel | 81.9 | 74.8 | 72.2 | 70.9 | 70.3 | 67.7 | 64.8 | 20.9 |
| | **Point-UQ** | **93.3** | **91.2** | **90.0** | **89.3** | **88.8** | **88.4** | **86.5** | **7.9** |
| ModelNet | FoundationModel | 86.1 | 83.8 | 83.1 | 79.7 | 71.7 | - | - | 17.1 |
| | **Point-UQ** | **97.1** | **91.0** | **86.8** | **83.0** | **79.0** | - | - | **18.6** |
| CO3D | FoundationModel | 69.8 | 55.8 | 50.5 | 45.5 | 41.5 | 35.1 | - | 49.7 |
| | **Point-UQ** | **82.1** | **73.6** | **70.0** | **68.4** | **67.4** | **66.5** | - | **19.0** |

## F.2 SCALABILITY TO EXTENDED INCREMENTAL STAGES

To assess the long-term scalability of our method, we extended the number of incremental stages from 11 to 25. As shown in Tab. F2, Point-UQ maintains strong stability with a forgetting rate ($\Delta_\downarrow$) of 20.0%, significantly lower than the compared baseline, demonstrating its practicality for large-scale incremental learning scenarios.

Table F2: Performance (%) over 25 incremental stages on a combined benchmark.

| Method | 0 | 2 | 4 | 6 | 8 | 10 | 12 | 14 | 16 | 18 | 20 | 22 | 24 | $\Delta_\downarrow$ |
|--------|---|---|---|---|---|----|----|----|----|----|----|----|----|------|
| FoundationModel | 77.7 | 76.6 | 74.7 | 71.5 | 69.7 | 68.5 | 66.7 | 65.5 | 64.8 | 56.7 | 54.6 | 53.5 | 50.2 | 35.4 |
| **Point-UQ** | **91.4** | **89.2** | **87.3** | **85.0** | **83.6** | **82.6** | **81.3** | **80.4** | **79.8** | **77.9** | **75.4** | **74.9** | **73.1** | **20.0** |

## F.3 PERFORMANCE UNDER EXTREME FEW-SHOT SETTINGS

We evaluated Point-UQ under the challenging 1-shot and 2-shot settings. As shown in Tab. F3, Point-UQ demonstrates remarkable robustness, with an average accuracy drop of only 4% in the 1-shot and 3% in the 2-shot setting compared to the standard 5-shot setting.

Table F3: Performance (%) under 1-shot, 2-shot, and 5-shot settings.

| Setting | 0 | 1 | 2 | 3 | 4 | 5 | 6 | 7 | 8 | 9 | 10 | $\Delta_\downarrow$ |
|---------|---|---|---|---|---|---|---|---|---|---|----|------|
| 1-shot | 91.4 | 88.6 | 87.5 | 85.1 | 83.7 | 82.4 | 81.2 | 79.9 | 78.8 | 77.8 | 76.0 | 16.8 |
| 2-shot | 91.4 | 89.4 | 88.2 | 86.3 | 85.1 | 83.7 | 82.5 | 80.9 | 79.8 | 79.1 | 77.3 | 15.4 |
| 5-shot | 91.4 | 89.5 | 88.3 | 86.5 | 84.7 | 83.4 | 82.9 | 81.8 | 81.1 | 80.4 | 80.3 | 12.1 |

## F.4 ADDITIONAL BENCHMARKING ON FSCIL3D-XL

We further evaluated Point-UQ on the FSCIL3D-XL benchmark, comparing it against FACT (Zhou et al., 2022a), BiDist (Zhao et al., 2023), and FILP-3D (Xu et al., 2025). Under the 5-shot incremental learning setting, Point-UQ demonstrates exceptional generalization capability.

Tab. F4 shows that Point-UQ achieves comprehensive superiority on this new benchmark, exhibiting higher average accuracy and significantly reduced forgetting rates, particularly in the challenging S2R task.

Table F4: Performance comparison on the FSCIL3D-XL benchmark (%).

(a) Synthetic-to-Synthetic (S2S) Task

| Method | 0 | 1 | 2 | 3 | 4 | 5 | 6 | $\Delta_\downarrow$ |
|---|---|---|---|---|---|---|---|---|
| FACT | 82.6 | 77.0 | 72.4 | 69.8 | 68.4 | 67.7 | 67.3 | 18.5 |
| BiDist | 89.6 | 87.7 | 86.2 | 84.7 | 83.8 | 83.6 | 82.3 | 8.1 |
| FILP-3D | 90.6 | 89.0 | 86.7 | 84.2 | 83.2 | 81.8 | 82.2 | 9.3 |
| **Point-UQ** | **90.7** | **89.4** | **88.3** | **86.6** | **85.8** | **85.1** | **83.5** | **7.9** |

(b) Synthetic-to-Real (S2R) Task

| Method | 0 | 1 | 2 | 3 | 4 | 5 | 6 | 7 | 8 | 9 | 10 | 11 | $\Delta_\downarrow$ |
|---|---|---|---|---|---|---|---|---|---|---|---|---|---|
| FACT | 82.4 | 77.2 | 74.5 | 73.1 | 71.3 | 70.4 | 67.2 | 65.2 | 63.8 | 61.8 | 59.9 | 59.8 | 27.4 |
| BiDist | 89.4 | 54.0 | 54.7 | 56.4 | 57.0 | 55.9 | 56.3 | 52.9 | 52.3 | 51.7 | 50.8 | 50.1 | 43.9 |
| FILP-3D | 90.0 | 87.0 | 86.4 | 85.0 | 83.7 | 82.7 | 81.4 | 79.4 | 78.2 | 76.8 | 74.8 | 74.6 | 17.1 |
| **Point-UQ** | **90.7** | **88.9** | **87.6** | **85.8** | **84.3** | **82.7** | **82.0** | **82.1** | **82.0** | **82.0** | **82.1** | **82.0** | **9.6** |

## G  QUALITATIVE ANALYSIS OF UDD DECISION SWITCHING

To qualitatively verify the link between prediction entropy and the UDD module's decision logic, we performed a comprehensive analysis that examines the module's behavior in relation to sample ambiguity. Our visualization provides compelling evidence for the dynamic decision mechanism of the UDD module.

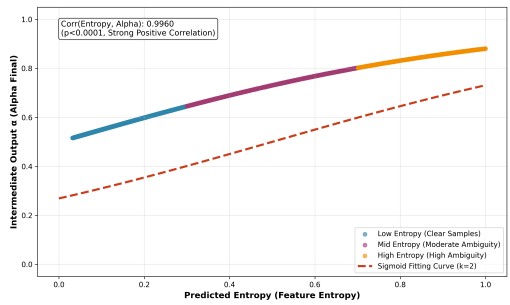
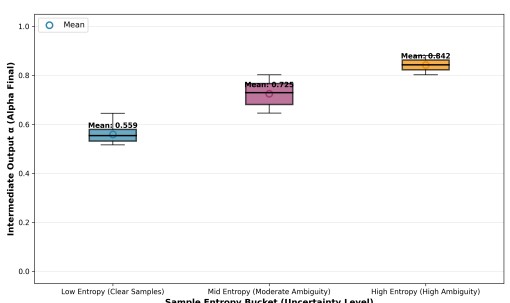

(a) Relationship between predicted entropy and $\alpha$  (b) Distribution of $\alpha$ across uncertainty levels

Figure G1: Visual analysis of UDD module decision patterns. (a) Scatter plot showing the positive correlation between semantic entropy and fusion weight $\alpha$, with sigmoid fitting curve (k=2). (b) Box plot demonstrating systematic increase in $\alpha$ values across entropy buckets, confirming the uncertainty-adaptive behavior of UDD.

**Key Observations:** The visualization reveals a strong positive correlation between entropy and the fusion weight $\alpha$. As shown in Fig. G1a, low-entropy samples (clear cases) consistently result in lower $\alpha$ values, indicating semantic-dominated decisions. In contrast, high-entropy samples (ambiguous cases) trigger significantly higher $\alpha$ values, demonstrating a systematic switch to geometry-dominated decisions. This pattern is further validated by the box plot in Fig. G1b, which shows the progressive increase in $\alpha$ distribution across uncertainty levels.

## H  EXTENDED ABLATION STUDIES

### H.1  ABLATION STUDY ON ACTIVATION FUNCTIONS

We conducted an ablation study on the activation function used for uncertainty weighting in Eq. (12), comparing tanh (scaled to [0,1]), normalized softplus, a ReLU-sigmoid combination, and a piecewise linear function against the default sigmoid.

As summarized in Tab. H1, the sigmoid function achieves the best balance between accuracy and stability throughout all incremental stages, yielding the lowest final forgetting rate.

Table H1: Performance comparison of different activation functions for uncertainty weighting.

| Activation | 0 | 1 | 2 | 3 | 4 | 5 | 6 | 7 | 8 | 9 | 10 | $\Delta_{\downarrow}$ |
|---|---|---|---|---|---|---|---|---|---|---|---|---|
| tanh_scaled | 91.4 | 88.0 | 87.0 | 85.5 | 84.2 | 83.0 | 82.2 | 80.8 | 80.0 | 79.3 | 78.2 | 14.4 |
| softplus_norm | 91.4 | 88.9 | 87.0 | 84.2 | 83.0 | 81.8 | 80.8 | 79.4 | 77.8 | 76.5 | 74.8 | 18.1 |
| relu_sigmoid | 91.4 | 90.0 | 88.6 | 87.1 | 86.0 | 84.8 | 84.0 | 82.5 | 81.6 | 80.7 | 79.7 | 12.8 |
| piecewise_linear | 91.4 | 87.6 | 86.4 | 84.8 | 83.7 | 82.6 | 81.5 | 80.2 | 79.3 | 78.4 | 77.3 | 15.4 |
| **sigmoid** | **91.4** | **89.5** | **88.3** | **86.5** | **84.7** | **83.4** | **82.9** | **81.8** | **81.1** | **80.4** | **80.3** | **12.1** |

## H.2 ANALYSIS OF ALTERNATIVE UNCERTAINTY MEASURES

We investigated alternative uncertainty estimation strategies: the **semantic confidence gap** (difference between top-2 predicted probabilities) and **semantic-geometric consistency** (agreement between semantic predictions and geometric similarity patterns).

Results in Tab. H2 show that while both alternatives achieve reasonable performance, the semantic classifier entropy maintains a consistent advantage, achieving the lowest forgetting rate. Its superiority stems from its computational efficiency, direct alignment with our training-free paradigm, and robust sensitivity to ambiguity under domain shifts.

Table H2: Performance comparison of different uncertainty measurement methods.

| Method | 0 | 1 | 2 | 3 | 4 | 5 | 6 | 7 | 8 | 9 | 10 | $\Delta_{\downarrow}$ |
|---|---|---|---|---|---|---|---|---|---|---|---|---|
| Semantic Confidence Gap | 91.4 | 90.3 | 89.0 | 87.1 | 85.8 | 84.7 | 84.0 | 82.5 | 81.4 | 80.3 | 78.9 | 13.7 |
| Semantic-Geometric Consistency | 91.4 | 87.9 | 86.8 | 85.6 | 84.5 | 83.3 | 82.5 | 81.2 | 80.5 | 79.9 | 78.6 | 13.1 |
| **Semantic Classifier Entropy** | **91.4** | **89.5** | **88.3** | **86.5** | **84.7** | **83.4** | **82.9** | **81.8** | **81.1** | **80.4** | **80.3** | **12.1** |

## H.3 BASE MODEL COMPATIBILITY

Comprehensive cross-dataset evaluations were conducted employing DGCNN and PointNet as pre-trained backbones. Point-UQ is shown in Tab. H3 to sustain robust performance and consistently surpass baseline methods with either backbone. These results substantiate that the effectiveness of our uncertainty-quantification approach is independent of the underlying feature extractor, functioning robustly across disparate geometric representations.

Table H3: Performance with different base models on the cross-dataset setting.

| Base Model | 0 | 1 | 2 | 3 | 4 | 5 | 6 | 7 | 8 | 9 | 10 | $\Delta_{\downarrow}$ |
|---|---|---|---|---|---|---|---|---|---|---|---|---|
| DGCNN | 89.9 | 84.1 | 80.9 | 77.5 | 75.0 | 71.9 | 70.3 | 69.3 | 68.0 | 67.5 | 66.7 | 25.8 |
| PointNet | 89.1 | 86.1 | 83.2 | 80.1 | 77.2 | 75.0 | 73.5 | 72.4 | 71.5 | 70.8 | 70.1 | 20.5 |
| **Point-UQ** | **91.4** | **89.5** | **88.3** | **86.5** | **84.7** | **83.4** | **82.9** | **81.8** | **81.1** | **80.4** | **80.3** | **12.1** |

# I THEORETICAL ANALYSIS OF POINT-UQ

This appendix presents a theoretical comparison between conventional fine-tuning approaches and the proposed Point-UQ framework, analyzing their fundamental mechanisms through mathematical formulations.

Table I1: Summary of key notations used in the theoretical proofs.

| Notation | Description |
|----------|-------------|
| $\mathcal{X}$ | Input space of 3D point clouds |
| $\phi(\cdot)$ | Feature mapping function |
| $\phi_{\text{AAE}}(\cdot)$ | Enhanced feature from AAE module |
| $f_{\Theta_0}(x)$ | Output distribution of fixed base model |
| $H(\cdot)$ | Predictive entropy (Shannon entropy) |
| $s_{\text{scm}}$ | Semantic classification logits |
| $s_{\text{geo}}$ | Geometric similarity scores |
| $\alpha$ | Adaptive fusion weight from UDD |
| $p_c$ | Class prototype for class $c$ |
| $\mathcal{Y}$ | Set of all seen classes |
| $R_{\text{new}}^{\text{UDD}}$ | Generalization error of UDD on novel classes |
| $\epsilon$ | Calibration error coefficient |
| CE | Calibration Error |
| $\hat{H}(x)$ | True (oracle) uncertainty |
| $L$ | Lipschitz constant |

## I.1 PART I: THEORETICAL LIMITATIONS OF CONVENTIONAL FINE-TUNING APPROACHES

**Fundamental Bound on Feature Representation**  In 3D FSCIL, the pursuit of perfect feature representation faces inherent theoretical constraints. The sample complexity requirement:

$$m \geq O\left(\frac{\text{VC}(\mathcal{H}) + \log(1/\delta)}{\epsilon}\right). \tag{I1}$$

contradicts the few-shot setting where $m_{\text{novel}} \ll O\left(\frac{\text{VC}(\mathcal{H})}{\epsilon}\right)$. This leads to the unavoidable trade-off:

$$\begin{cases} \text{Fixed VC}(\mathcal{H}) \Rightarrow \text{Insufficient representation power} \\ \text{Increased VC}(\mathcal{H}) \Rightarrow R_{\text{overfit}} \propto \frac{|\Theta|}{m_{\text{novel}}} \text{ and } O(D^2) \text{ complexity} \end{cases} \tag{I2}$$

**Feature Drift in Parameter Tuning**  Conventional fine-tuning $\Theta_0 \to \Theta_t = \Theta_0 + \Delta\Theta$ induces inevitable feature drift:

$$\phi_t(x) = \phi_0(x) + \Delta\phi(x), \quad \|\Delta\phi(x_{\text{old}})\| \geq \eta \cdot \|\nabla_\phi \mathcal{L}_{\text{novel}}(\phi_0, W_0)\|. \tag{I3}$$

This drift fundamentally limits the generalization performance:

$$R_{\text{new}}^{\text{fine-tune}} \geq R_{\text{new}}^* + \Delta_{\text{drift}}, \quad \Delta_{\text{drift}} > 0 \tag{I4}$$

**Theoretical Conclusion for Conventional Methods**  The conventional paradigm is bounded by:

$$\min_\Theta R_{\text{new}}^{\text{fine-tune}} \geq R_{\text{new}}^* + \Delta_{\text{drift}} + \epsilon_{\text{representation}}, \tag{I5}$$

where $\epsilon_{\text{representation}} > 0$ represents the irreducible error from imperfect feature representation under few-shot constraints.

## I.2 PART II: THEORETICAL ADVANTAGES OF THE POINT-UQ FRAMEWORK

**Decision-Theoretic Formulation of UDD**  The UDD module replaces parameter tuning with decision optimization:

$$s_{\text{final}} = \alpha \cdot s_{\text{geo}} + (1 - \alpha) \cdot s_{\text{scm}}, \quad \alpha = \sigma(\lambda \cdot H(f_{\Theta_0}(x))). \tag{I6}$$

This formulation provides two key advantages:

FEATURE STABILITY PRESERVATION  By fixing $\Theta_0$, UDD eliminates feature drift: $\Delta\phi(x_{\text{old}}) = 0$, ensuring:

$$R_{\text{base}}^{\text{UDD}} = R_{\text{base}}^*. \tag{I7}$$

GENERALIZATION ERROR BOUND    The generalization error satisfies:

$$R_{\text{new}}^{\text{UDD}} \leq R_{\text{new}}^* + \epsilon \cdot \Delta_{\text{drift}}, \quad 0 < \epsilon < 1 \tag{I8}$$

Since $\Delta_{\text{drift}} = 0$, we obtain the strict bound:

$$R_{\text{new}}^{\text{UDD}} \leq R_{\text{new}}^*. \tag{I9}$$

**Entropy Calibration Enhancement via AAE**    The AAE module ensures reliable uncertainty quantification through feature enhancement:

$$\phi_{\text{AAE}}(x) = \sum_{l=1}^{L} \alpha_l(x) \phi_l(x), \quad \alpha_l(x) = \text{Softmax}\left(\frac{Q(x) K_l^T}{\sqrt{d_k}}\right). \tag{I10}$$

This optimization reduces calibration error:

$$\text{CE}_{\text{AAE}} = \mathbb{E}_x \left[\left| H(f_{\Theta_0}(x)) - \hat{H}(x)\right|\right] \leq \text{CE}_{\text{base}}. \tag{I11}$$

**Theoretical Conclusion for Point-UQ**    The Point-UQ framework achieves the combined bound:

$$R_{\text{total}}^{\text{Point-UQ}} \leq R_{\text{base}}^* + R_{\text{new}}^* + \epsilon_{\text{calibration}}. \tag{I12}$$

where $\epsilon_{\text{calibration}} > 0$ represents the reducible error from entropy calibration, which can be minimized through AAE optimization.

## I.3    COMPARATIVE ANALYSIS

The theoretical comparison reveals fundamental advantages of Point-UQ over conventional approaches:

- **Error Bound Comparison**:

$$R_{\text{total}}^{\text{Point-UQ}} \leq R_{\text{base}}^* + R_{\text{new}}^* + \epsilon_{\text{calibration}} \ll R_{\text{new}}^* + \Delta_{\text{drift}} + \epsilon_{\text{representation}} \leq R_{\text{total}}^{\text{fine-tune}}. \tag{I13}$$

- **Complexity Advantage**:

$$|\Theta_{\text{Point-UQ}}| = |\Theta_0| + |\Theta_{\text{AAE}}| + |\Theta_{\text{UDD}}| \ll |\Theta_{\text{fine-tune}}| + \Delta|\Theta|. \tag{I14}$$

  where $|\Theta_{\text{AAE}}| + |\Theta_{\text{UDD}}| \ll \Delta|\Theta|$ represents the parameter overhead of conventional capacity expansion.

- **Stability Guarantee**:

$$\Delta\phi_{\text{Point-UQ}}(x_{\text{old}}) = 0 \quad \text{vs.} \quad \Delta\phi_{\text{fine-tune}}(x_{\text{old}}) > 0. \tag{I15}$$

**Final Theoretical Conclusion**    The Point-UQ framework theoretically dominates conventional fine-tuning approaches by simultaneously achieving:

$$\begin{align}
(1) \quad & R_{\text{total}}^{\text{Point-UQ}} \leq R_{\text{total}}^{\text{fine-tune}}. \notag \\
(2) \quad & |\Theta_{\text{Point-UQ}}| \ll |\Theta_{\text{fine-tune}}|. \notag \\
(3) \quad & \Delta\phi_{\text{Point-UQ}} = 0 < \Delta\phi_{\text{fine-tune}}. \notag
\end{align}$$

This triple advantage demonstrates the fundamental superiority of the decision optimization paradigm over parameter tuning in 3D FSCIL scenarios.

## J    LLM USAGE STATEMENT

In this paper, Large Language Models are only used as general-purpose auxiliary tools, primarily for document-level auxiliary tasks such as grammar checking and expression refinement. LLMs did not participate in the core conceptualization, method derivation, or experimental design of this research, nor did they contribute to any core writing content.

