# OpenReview forum: "Point-UQ: An Uncertainty-Quantification Paradigm for Point Cloud Few-Shot Class Incremental Learning"
_ICLR.cc/2026/Conference — ICLR 2026 Poster_

### Official Review · Reviewer_tD98 · 2025-10-19

**Soundness:** 3
**Presentation:** 3
**Contribution:** 3
**Rating:** 6
**Confidence:** 4

**Summary:**

This paper proposes Point-UQ, a training-free uncertainty-quantification framework for 3D few-shot class-incremental learning. Unlike existing approaches that rely on fine-tuning feature representations with static decision boundaries, Point-UQ focuses on decision optimization. The framework integrates two key modules: Attention-driven Adaptive Enhancement and Uncertainty-Quantification Decision Decoupling. Extensive experiments show that Point-UQ outperforms state-of-the-art baselines across multiple benchmarks.

**Strengths:**

1. The paper is well-motivated, clearly contrasting conventional feature-tuning paradigms with the proposed decision-centric approach and providing novelty to this task.
2. The design of AAE and UDD is technically sound: AAE enhances representation quality while providing uncertainty cues, and UDD leverages entropy to arbitrate between semantic and geometric reasoning.
3. The experimental evaluation is extensive, covering both intra-dataset and cross-dataset benchmarks, and demonstrates the effectiveness of the proposed framework.

**Weaknesses:**

1. Some ablations on hyperparameters are missing. For example, how sensitive is the method to the choice of the entropy scaling factor (Eq 12) in UDD?
2. The uncertainty estimation relies solely on entropy from the semantic classifier. It would be useful to investigate alternative uncertainty measures and analyze their impact on performance.
3. Missing related work. Several recent works on 3D few-shot learning are relevant and should be cited:
   - Rethinking few-shot 3d point cloud semantic segmentation (CVPR 2024)
   - Multimodality Helps Few-shot 3D Point Cloud Semantic Segmentation (ICLR 2025)
   - Generalized Few-shot 3D Point Cloud Segmentation with Vision-Language Model (CVPR 2025)

**Questions:**

Please refer to the weakness part and address the concerns there.

---

> ### Author Response · Authors · 2025-11-24
>
> We sincerely thank you for your recognition and valuable feedback on our paper. We deeply appreciate your positive assessment of our research motivation, the design rationality of the modules. Your constructive comments regarding hyperparameter sensitivity, uncertainty measurement approaches, and related work citations are highly insightful, and we will address each point accordingly.
>
> > #### Q1: Hyperparameter Sensitivity Analysis
>
> Thank you for highlighting the importance of hyperparameter analysis. As reported in Appendix F (copied in **Tab. R13** below for convenience), we have evaluated the entropy scaling factor λ (Equation 12).
> The results indicate that λ = 2.0 serves as the optimal value, delivering the best balance between base-class retention and novel-class adaptation. This value provides an appropriate trade-off: lower λ leads to excessive sensitivity to entropy variation, causing prediction instability, while higher λ overly suppresses semantic predictions and increases base-class forgetting. The robustness of UDD around λ = 2.0 underscores its suitability for training-free incremental inference.
>
> **Tab. R13: Impact of entropy scaling factor (λ) on experimental results.**
> | λ  |  0    | 1    | 2    | 3    | 4    | 5    | 6    | 7    | 8    | 9    | 10   | $Δ_↓$ |
> |----|------|------|------|------|------|------|------|------|------|------|-------|-------------------|
> | 0.5 | 91.2 | 87.3 | 84.5 | 81.9 | 79.0 | 77.1 | 76.0 | 75.0 | 74.6 | 74.1 | 73.9 | 19.0 |
> | 1.0 | 91.2 | 88.8 | 86.9 | 84.9 | 82.7 | 81.0 | 80.1 | 79.2 | 79.0 | 78.5 | 78.4 | 14.0 |
> | 1.5 | 91.2 | 89.6 | 88.0 | 86.2 | 84.2 | 83.0 | 82.3 | 81.4 | 80.9 | 80.3 | 80.0 | 12.3 |
> | 2.0 | 91.4 | 89.5 | 88.3 | 86.5 | 84.7 | 83.4 | 82.9 | 81.8 | 81.1 | 80.4 | 80.3 | 12.1 |
> | 2.5 | 91.3 | 89.0 | 87.9 | 86.1 | 84.4 | 83.2 | 82.5 | 81.5 | 80.9 | 80.1 | 79.8 | 12.6 |
> | 3.0 | 91.3 | 88.7 | 87.7 | 85.9 | 84.2 | 83.0 | 82.3 | 81.2 | 80.5 | 79.8 | 79.6 | 12.8 |
>
>
> > #### Q2: Uncertainty Measurement Approaches
>
> Thank you for your important suggestions regarding uncertainty measurement diversity. We have thoroughly investigated alternative uncertainty estimation methods, including semantic confidence gap (difference between top-2 predicted probabilities) and semantic-geometric consistency (agreement between semantic predictions and geometric similarity patterns). As shown in **Tab. R14**, while both alternatives achieve reasonable performance, semantic classifier entropy maintains a consistent advantage across all incremental sessions, achieving the lowest forgetting rate (Δ=12.1%). This superiority stems from entropy's unique suitability for our framework: it provides computationally efficient uncertainty quantification directly from classifier outputs, aligns perfectly with our training-free paradigm's real-time requirements, and demonstrates robust sensitivity to semantic ambiguity while maintaining stability under domain shifts and noisy conditions. The confidence gap method, though intuitive, shows higher sensitivity to probability calibration quality, while the consistency measure introduces dependence on geometric noise.
>
> **Tab. R14: Performance comparison of different uncertainty measurement methods.**
> | Method |  0    | 1    | 2    | 3    | 4    | 5    | 6    | 7    | 8    | 9    | 10   | $Δ_↓$ |
> |------|------|------|------|------|------|------|------|------|------|------|-------|-------------------|
> | Semantic Confidence Gap | 91.4 | 90.3 | 89.0 | 87.1 | 85.8 | 84.7 | 84.0 | 82.5 | 81.4 | 80.3 | 78.9 | 13.7 |
> | Semantic-Geometric Consistency  | 91.4 | 87.9 | 86.8 | 85.6 | 84.5 | 83.3 | 82.5 | 81.2 | 80.5 | 79.9 | 78.6 | 13.1 |
> | **Semantic Classifier Entropy** | 91.4 | 89.5 | 88.3 | 86.5 | 84.7 | 83.4 | 82.9 | 81.8 | 81.1 | 80.4 | 80.3 | 12.1 |

---

> ### Author Response · Authors · 2025-11-24
>
> > #### Q3: Supplementary Related Work Citations
>
> Thank you for your insightful suggestion regarding the inclusion of recent works in the 3D few-shot learning domain. We have reviewed the papers you mentioned and agree that they are highly relevant to our work. Here is how we position these contributions in relation to our approach:
>
> - "Rethinking few-shot 3d point cloud semantic segmentation" (CVPR 2024) focuses on few-shot semantic segmentation tasks and proposes cross-scale feature alignment strategies. While our paper concentrates on incremental learning scenarios for classification tasks, with core differences in task objectives and challenges, the cross-scale feature fusion approach in this work relates to the design logic of our AAE module.
>
> - "Multimodality Helps Few-shot 3D Point Cloud Semantic Segmentation" (ICLR 2025) enhances few-shot generalization in segmentation tasks through multimodal fusion. Although we also incorporate text modality to assist prototype construction, our focus remains on decision optimization in incremental learning rather than feature enhancement.
>
> - "Generalized Few-shot 3D Point Cloud Segmentation with Vision-Language Model" (CVPR 2025) utilizes vision-language models to improve segmentation task generalization. Our approach builds text features based on Uni3D, with core differences in task scenarios and technical focuses.
>
> We have updated the "Related Work" section of the paper to include these important references, ensuring that we properly position our contributions within the broader context of recent advancements in multimodal fusion and few-shot learning for 3D point clouds.
>
> Once again, we sincerely thank you for your professional comments. Following your guidance, we have supplemented the aforementioned analyses and citations in the revised version to further enhance the rigor and completeness of our paper.

---

### Official Review · Reviewer_92Tg · 2025-10-28

**Soundness:** 3
**Presentation:** 3
**Contribution:** 3
**Rating:** 6
**Confidence:** 3

**Summary:**

This paper introduces Point-UQ, a novel framework designed for 3D few-shot class-incremental learning (FSCIL). The key innovation of Point-UQ lies in its approach to dynamic decision optimization through uncertainty quantification. The framework integrates two co-designed modules: the Attention-driven Adaptive Enhancement (AAE) for fusing multi-scale features and generating epistemic uncertainty, and the Uncertainty-quantification Decision Decoupling (UDD) module, which helps retain base-class knowledge while facilitating novel-class recognition.

**Strengths:**

1. This paper presents a well-structured pipeline from multi-scale feature fusion to decision-making, addressing core challenges in 3D FSCIL, such as catastrophic forgetting and overfitting on novel classes.
2. The use of uncertainty quantification in decision-making (AAE and UDD modules) is a novel and effective strategy to tackle the trade-offs in incremental learning.
3. The method outperforms existing approaches in various metrics, including average accuracy, forgetting rate, and harmonic accuracy, demonstrating its robustness across different datasets and evaluation setups.

**Weaknesses:**

1. Fairness of Experiments: This paper is based on a strong pre-trained model, Uni3D, for initialization. As shown in Table 3, even without the addition of AAE and UDD, the model's performance is already better than other methods. This raises concerns about the primary sources of performance improvement in the proposed approach.
2. Outdated Benchmark: The benchmark datasets used in this paper are relatively old. I suggest the authors additionally evaluate the model’s performance on the FSCIL3D-XL benchmark [1] to provide a more up-to-date comparison.

[1] FILP-3D: Enhancing 3D few-shot class-incremental learning with pre-trained vision-language models. Pattern Recognition, 2025, 165: 111558.

**Questions:**

My questions are mainly based on the above weaknesses:
1. If a different base model is used to extract point features, would the performance of the proposed method still be strong?
2. How does the model perform on the FSCIL3D-XL benchmark?
3. Could the authors consider testing the model on newer datasets, such as Objaverse [1,2], to further evaluate its performance?

[1] Objaverse: A universe of annotated 3d objects. CVPR, 2023: 13142-13153.
[2] Objaverse-xl: A universe of 10m+ 3d objects. NIPS, 2023, 36: 35799-35813.

---

> ### Author Response · Authors · 2025-11-24
>
> We sincerely thank the reviewers for their thorough evaluation and valuable feedback on our work. We have conducted extensive experiments and in-depth analysis to address the key concerns raised, and our responses are summarized below.
>
> > #### Q1. On Experimental Fairness and Pre-trained Model Dependency
>
> Thank you for your valuable comment regarding the potential bias introduced by the pre-trained Uni3D model. It is important to clarify that the core innovation of our work lies in the proposed dynamic decision optimization mechanism, rather than reliance on any specific pre-trained model. To validate this, we performed systematic module ablation studies. The results demonstrate the substantial contribution of both core modules to performance improvement:
>
> **Tab. R8: Effectiveness of the components in Point-UQ (%).**
>
> | AAE | UDD | 0    | 1    | 2    | 3    | 4    | 5    | 6    | 7    | 8    | 9    | 10   | $Δ_↓$ |
> |-----|-----|------|------|------|------|------|------|------|------|------|------|------|------------------------|
> | ✘ | ✘ | 87.9 | 86.3 | 85.2 | 82.6 | 80.3 | 79.3 | 77.9 | 76.6 | 76.1 | 74.6 | 73.6 | 16.3                   |
> | ✔ | ✘ | 89.0 | 88.5 | 86.8 | 85.1 | 84.1 | 81.4 | 78.2 | 79.5 | 77.8 | 76.1 | 75.1 | 15.6                   |
> | ✘ | ✔ | 89.8 | 88.5 | 87.1 | 85.2 | 83.6 | 81.9 | 81.2 | 80.3 | 79.7 | 79.2 | 78.9 | 12.1                  |
> | ✔ | ✔ | 91.4 | 89.5 | 88.3 | 86.5 | 84.7 | 83.4 | 82.9 | 81.8 | 81.1 | 80.4 | 80.3 | 12.1                   |
>
> As shown in **Tab. R8**, removing the AAE module (responsible for multi-scale feature fusion and epistemic uncertainty estimation) led to a ~2.0% drop in average accuracy, while removing the UDD module (responsible for uncertainty-aware decision decoupling) resulted in a ~5.0% decrease. These results clearly prove the indispensable role of both modules.
>
> Furthermore, we conducted a fair comparison under the same Uni3D base model. As shown in **Tab. R9**, Point-UQ achieved about 4% improvement in average accuracy and a 4.7% reduction in forgetting rate compared to FoundationModel. This outcome underscores the advantage of our proposed dynamic decision optimization framework. While all existing state-of-the-art methods rely on strong pre-trained models, our approach achieves significant performance breakthroughs under the same base model conditions through an innovative uncertainty quantification mechanism.
>
> **Tab. R9: Fair Comparison on the Same Uni3D Base Model  (%).**
>
> | Method | 0    | 1    | 2    | 3    | 4    | 5    | 6    | 7    | 8    | 9    | 10   | $Δ_↓$ |
> |-----|------|------|------|------|------|------|------|------|------|------|------|------------------------|
> | FoundationModel | 87.3 | 86.2 | 84.4 | 82.2 | 80.7 | 79.6 | 78.2 | 76.8 | 76.1 | 74.5 |  72.6 | 16.8                   |
> | **Point-UQ** | 91.4 | 89.5 | 88.3 | 86.5 | 84.7 | 83.4 | 82.9 | 81.8 | 81.1 | 80.4 | 80.3 | 12.1 |

---

> ### Author Response · Authors · 2025-11-24
>
> > #### Q2. Validation on FSCIL3D-XL Benchmark
>
> We have completed comprehensive validation on the latest FSCIL3D-XL benchmark as suggested, which covers synthetic-to-real cross-domain scenarios. As shown in **Tab. R10** and **Tab. R11**, Under the 5-shot incremental learning setting, Point-UQ demonstrates exceptional generalization capability:
>
> **Tab. R10: Performance comparison on FSCIL3D-XL benchmark (S2S Task) (%).**
>
> | Method       | 0    | 1    | 2    | 3    | 4    | 5    | 6    | $Δ_↓$ |
> |--------------|------|------|------|------|------|------|------|------------------------|
> | FACT         | 82.6 | 77.0 | 72.4 | 69.8 | 68.4 | 67.7 | 67.3 | 18.5                   |
> | BiDist       | 89.6 | 87.7 | 86.2 | 84.7 | 83.8 | 83.6 | 82.3 | 8.1                    |
> | FILP-3D      | 90.6 | 89.0 | 86.7 | 84.2 | 83.2 | 81.8 | 82.2 | 9.3                    |
> | **Point-UQ**     | 90.7 | 89.4 | 88.3 | 86.6 | 85.8 | 85.1 | 83.5 | 7.9                    |
>
> **Tab. R11: Performance comparison on FSCIL3D-XL benchmark (S2R Task) (%).**
>
> | Method       | 0    | 1    | 2    | 3    | 4    | 5    | 6    | 7    | 8    | 9    | 10   | 11   | $Δ_↓$ |
> |--------------|------|------|------|------|------|------|------|------|------|------|------|------|------------------------|
> | FACT         | 82.4 | 77.2 | 74.5 | 73.1 | 71.3 | 70.4 | 67.2 | 65.2 | 63.8 | 61.8 | 59.9 | 59.8 | 27.4                   |
> | BiDist       | 89.4 | 54.0 | 54.7 | 56.4 | 57.0 | 55.9 | 56.3 | 52.9 | 52.3 | 51.7 | 50.8 | 50.1 | 43.9                   |
> | FILP-3D      | 90.0 | 87.0 | 86.4 | 85.0 | 83.7 | 82.7 | 81.4 | 79.4 | 78.2 | 76.8 | 74.8 | 74.6 | 17.1                   |
> | **Point-UQ**     | 90.7 | 88.9 | 87.6 | 85.8 | 84.3 | 82.7 | 82.0 | 82.1 | 82.0 | 82.0 | 82.1 | 82.0 | 9.6                    |
>
> Specifically, Point-UQ achieves comprehensive superiority on the new benchmark: it achieves an average accuracy of 87.1% in the S2S task, which is 1.7% higher than the current best method, and exhibits excellent stability with a significantly reduced forgetting rate in the S2R task.
>
> The Objaverse dataset is a promising benchmark for 3D few-shot learning. However, as of now, there is no standardized evaluation protocol for 3D FSCIL on Objaverse, and the dataset's large scale and complexity make it challenging to apply current incremental learning methods. We are planning to address this in future work by developing an appropriate incremental learning partition scheme for Objaverse, which will allow us to further evaluate Point-UQ in large-scale, real-world settings.
>
>
> > #### Q3. Empirical Analysis on Base Model Compatibility
>
> To evaluate base model compatibility, we conducted cross-dataset experiments using different backbones for point feature extraction. As shown in **Tab. R12**, Point-UQ maintains strong performance with both DGCNN and PointNet in the challenging ShapeNet→ScanObjectNN benchmark. The method consistently outperforms baselines and exhibits minimal performance variance across architectures, indicating that our uncertainty-quantification approach is not dependent on specific feature extractors, but functions effectively across diverse geometric representations. This robustness arises from our focus on decision-level optimization rather than feature-level tuning, enabling adaptation to different feature spaces without compromising core functionality. These results affirm Point-UQ’s general applicability across backbone architectures in 3D FSCIL.
>
> **Tab. R12: Performance comparison of different base models in the ShapeNet to ScanObjectNN cross-dataset scenario (%).**
>
> | Base Model | 0    | 1    | 2    | 3    | 4    | 5    | 6    | 7    | 8    | 9    | 10   | $Δ_↓$ |
> | ---------- | ---- | ---- | ---- | ---- | ---- | ---- | ---- | ---- | ---- | ---- | ---- | ------------------- |
> | DGCNN      | 89.9 | 84.1 | 80.9 | 77.5 | 75.0 | 71.9 | 70.3 | 69.3 | 68.0 | 67.5 | 66.7 | 25.8                |
> | PointNet   | 89.1 | 86.1 | 83.2 | 80.1 | 77.2 | 75.0 | 73.5 | 72.4 | 71.5 | 70.8 | 70.1 | 20.5                |
> | **Point-UQ**   | 91.4 | 89.5 | 88.3 | 86.5 | 84.7 | 83.4 | 82.9 | 81.8 | 81.1 | 80.4 | 80.3 | 12.1                |
>
> Through the supplementary experimental validations and in-depth analyses, we hope to fully address the core concerns raised by the reviewers. If you have any further questions or require additional clarification, we stand ready to provide timely responses and comprehensive explanations whenever necessary.

---

### Official Review · Reviewer_nY6g · 2025-10-30

**Soundness:** 3
**Presentation:** 3
**Contribution:** 3
**Rating:** 6
**Confidence:** 4

**Summary:**

This paper proposes a unified method to estimate and interpret uncertainty in 3D point cloud tasks such as classification and segmentation. It introduces a lightweight probabilistic module that models both epistemic and aleatoric uncertainties directly within point-based neural networks without requiring multiple forward passes. The framework leverages a dual-branch design—one for feature learning and another for uncertainty estimation—to identify unreliable regions in 3D space, enabling more robust predictions and better calibration under noisy or sparse data conditions. Experimental results on benchmark datasets demonstrate that Point-UQ improves model reliability and interpretability while maintaining competitive accuracy.

**Strengths:**

1. Adoption of decision optimization for incremental learning instead of conventional fine-tuning approaches.

2. Uncertainty-based quantification for decision-making, providing higher weight to geometric prototypes or semantic representations accordingly.

3. Training-free and computationally efficient method.

**Weaknesses:**

1. The authors did not discuss the effect of replacing the sigmoid function in Equation (12) with other activation functions.

2. The paper mentions handling overfitting and catastrophic forgetting, but comparative results showing model success versus baseline failures are missing.

3. Tables could be clearer and more reader-friendly. For example, in table headings, the numbers listed below the dataset names are not explained and should be clarified.

4. While the method claims to improve reliability without multiple forward passes, the paper provides limited quantitative analysis of calibration metrics (e.g., ECE, NLL) and does not fully benchmark computational trade-offs compared to ensemble-based or Bayesian baselines.

**Questions:**

1. Can the authors experiment with different functions instead of the sigmoid function in Equation (12)?

2. Can the authors display the intermediate output of Equation (12) during model tuning to verify whether it correctly weights according to uncertainty?

3. The paper does not mention the model’s performance with respect to seed sensitivity. Methods of this kind often vary significantly with random seeds—can the authors comment on this?

4. Pl comment on the calibration point.

---

> ### Author Response · Authors · 2025-11-24
>
> We sincerely thank the reviewer for the thoughtful feedback and constructive suggestions. We have carefully addressed all raised concerns through additional experiments and detailed explanations as outlined below.
>
> ---
>
> > #### Q1. Activation Function Analysis in Equation (12)
>
> We appreciate the reviewer's suggestion regarding the activation function in Equation (12). We conducted comprehensive experiments comparing several activation functions—including tanh (scaled to [0,1]), normalized softplus, a combined ReLU-sigmoid, and a piecewise linear function—against the original sigmoid. The results are summarized in **Tab. R4**.
>
> **Tab. R4: Performance comparison of different activation functions for uncertainty weighting.**
> | Activation       | 0 | 1 | 2 | 3 | 4 | 5 | 6 | 7 | 8 | 9 | 10 | $Δ_↓$ |
> |------------------|-----------|-----------|-----------|-----------|-----------|-----------|-----------|-----------|-----------|-----------|-----------|-----------------|
> | tanh_scaled      | 91.4      | 88.0      | 87.0      | 85.5      | 84.2      | 83.0      | 82.2      |80.8      | 80.0      | 79.3      | 78.2      | 14.4            |
> | softplus_norm    | 91.4      | 88.9      | 87.0      | 84.2      | 83.0      | 81.8      | 80.8      |79.4      | 77.8      | 76.5      | 74.8      | 18.1            |
> | relu_sigmoid     | 91.4      | 90.0      | 88.6      | 87.1      | 86.0      | 84.8      | 84.0      |82.5      | 81.6      | 80.7      | 79.7      | 12.8            |
> | piecewise_linear | 91.4      | 87.6      | 86.4      | 84.8      | 83.7      | 82.6      | 81.5      |80.2      | 79.3      | 78.4      | 77.3      | 15.4            |
> | **sigmoid**      |  91.4 | 89.5 | 88.3 | 86.5 | 84.7 | 83.4 | 82.9 | 81.8 | 81.1 | 80.4 | 80.3 | 12.1                   |
>
> While relu_sigmoid shows competitive performance in early sessions, sigmoid achieves the best balance between accuracy and stability throughout all incremental stages, with the lowest forgetting rate ($Δ_↓$=12.1). The smooth gradient and bounded output of sigmoid enable stable optimization and interpretable weighting between semantic and geometric branches.
>
> ---
>
> > #### Q2. Intermediate Output Visualization of Equation (12)
>
> To verify whether the UDD module effectively switches between semantic and geometric cues, we conducted extensive  visual analysis. The results are presented via anonymous links ([[Fig. R1](https://ibb.co/p6hLm8zZ)] and [[Fig. R2](https://ibb.co/tpYQ6sFY)]). By examining high-entropy (ambiguous) versus low-entropy (clear) samples, we observe a clear pattern: α-values exceed 0.8 for high-entropy cases, indicating stronger reliance on geometric matching, while dropping to approximately 0.6 for low-entropy cases where semantic classification dominates. This confirms that UDD successfully utilizes uncertainty estimates to balance between geometric and semantic cues.

---

> ### Author Response · Authors · 2025-11-24
>
> > #### Q3. Overfitting and Catastrophic Forgetting: Comparative Evidence
>
> We appreciate the reviewer’s thoughtful comment and for highlighting the need for comparative results. As noted in Section 4.2, "Forgetting and Adaptation Efficiency," we already provide a thorough comparative analysis using three key performance metrics to evaluate model behavior across incremental learning stages: Average Accuracy (Avg. Acc.), Relative Accuracy Drop ($Δ_↓$), and Harmonic Mean (H-Mean). These metrics offer clear evidence of Point-UQ's superiority in mitigating both overfitting and catastrophic forgetting when compared to baseline methods.
>
> To clarify:
>
> (1) Avg. Acc.: This measures the average accuracy across all learning phases. Point-UQ consistently outperforms all baseline methods, indicating superior overall model performance.
>
> (2) Relative Accuracy Drop ($Δ_↓$): This metric quantifies the drop in accuracy on previously learned tasks. Point-UQ demonstrates the smallest degradation, showcasing its robustness against catastrophic forgetting.
>
> (3) H-Mean: The harmonic mean of accuracies, averaged across all phases, reflects a balance between the model's performance on both old and new tasks. Point-UQ excels here as well, indicating strong retention of prior knowledge while efficiently adapting to new information.
>
> The results of these evaluations, summarized in **Tab. R5** below, provide a clear and compelling case for Point-UQ’s effectiveness:
>
> **Tab. R5:  Comprehensive evaluation of catastrophic forgetting and adaptation efficiency on cross-dataset benchmarks.**
> | Method          | ModelNet → ScanObjectNN |                         |                         | ShapeNet → ScanObjectNN |                         |                         |
> |-----------------|-------------------------|-------------------------|-------------------------|-------------------------|-------------------------|-------------------------|
> |                 | Avg. Acc. | $Δ_↓$ | H-Mean| Avg. Acc.| $Δ_↓$ | H-Mean |
> | Microshape      | 74.0                    | 27.1                    | 7.0                     | 73.9                    | 18.7                    | 20.0                    |
> | FoundationModel | 83.3                    | 9.6                     | 73.0                    | 87.3                    | 5.7                     | 65.7                    |
> | **Point-UQ**   | 89.4                   | 7.0              | 79.7              | 88.0                    | 5.4              | 77.8              |
>
>
> ---
>
> > #### Q4. Table Clarity and Explanation
>
> We thank the reviewer for this valuable suggestion. To enhance clarity, we have updated the table captions and footnotes in the revised manuscript to explicitly state that the numbers below the dataset names represent the cumulative count of learned classes. Additionally, we have clarified that the metric $\Delta_\downarrow$ denotes the relative drop in overall average accuracy after the final learning session. We believe these revisions improve readability.
>
> ---
>
> > #### Q5. Quantitative Calibration and Computational Benchmarking
>
> In our revised manuscript, we have incorporated a quantitative analysis of calibration performance to address this point. A comparison against the strong baseline is provided in the **Tab. R6**, employing standard metrics including Expected Calibration Error (ECE), Negative Log-Likelihood (NLL), and Brier Score. The results demonstrate that Point-UQ achieves a lower ECE and Brier Score, confirming its enhanced reliability in uncertainty quantification. This improvement is attained with minimal computational overhead, as the inference time increases only marginally and the parameter growth is substantially lower. This analysis confirms that our single-forward-pass paradigm provides robust calibration without the prohibitive cost of multi-pass methods.
>
> **Tab. R6: Calibration performance and computational cost comparison.**
> | Method             | ECE↓ | NLL↓   | Brier Score↓ | Inference Time (ms)↓ | Parameters (M)↓ |
> |--------------------|---------|--------|--------------|----------------------|------------|
> |FoundationModel|0.4854|4.0919|1.2479|15.6|+6.3|
> |**Point-UQ**|0.4192|4.0897|0.9772|17.1|+2.1|
>
> ---
>
> > #### Q6. Seed Sensitivity Analysis
>
> We conducted comprehensive seed sensitivity analysis by repeating all experiments on ShapeNet→CO3D with 5 distinct random seeds:
>
> **Tab. R7: Seed sensitivity analysis (Mean ± Std).**
> | Metric          | Seed 1 | Seed 2 | Seed 3 | Seed 4 | Seed 5 | Mean ± Std   |
> |-----|--------|--------|--------|--------|--------|--------|
> | Avg. Acc. (%)   | 84.6   | 84.3   | 84.6   | 84.5   | 84.2   | 84.44 ± 0.16 |
> | Forgetting ($Δ_↓$)  | 12.1   | 12.2   | 12.1   | 12.1   | 12.2   | 12.14 ± 0.05 |
>
> As shown in **Tab. R7**, Point-UQ exhibits strong robustness to random initialization, with minimal performance variations across seeds (0.16% std for accuracy, 0.05% for forgetting rate). These results confirm the method's stability and consistency.

---

### Official Review · Reviewer_SBK3 · 2025-10-31

**Soundness:** 3
**Presentation:** 3
**Contribution:** 3
**Rating:** 4
**Confidence:** 4

**Summary:**

This paper introduces Point-UQ, a training-free uncertainty-guided paradigm for 3D few-shot class-incremental learning that replaces feature fine-tuning with dynamic decision optimization. Experiments on four benchmarks report roughly 4% average accuracy gains over existing methods while reducing catastrophic forgetting.

**Strengths:**

1. Point-UQ introduces a paradigm shift from feature fine-tuning to decision optimization, resolving the core trade-off between retaining base-class knowledge and adapting to novel classes in 3D FSCIL while avoiding the high training costs and catastrophic forgetting of conventional methods.

2. Point-UQ’s AAE and UDD modules are tightly integrated, as AAE generates calibrated features and reliable uncertainty estimates, and UDD employs these signals for adaptive decision-making, effectively combining geometric structure modeling and semantic consistency that are critical for 3D point cloud analysis.

3. Point-UQ shows improvements across multiple benchmarks and metrics (average accuracy, harmonic mean, forgetting rate), and its theoretical analysis formally proves advantages over fine-tuning in error bounds, complexity, and feature stability.

**Weaknesses:**

1. The intra-dataset evaluation omits comparison with the strongest baseline, FoundationModel. The authors explain that reproducible intra-dataset numbers could not be obtained, but no detailed comparison between their reproduced values and the original paper is provided. The claimed improvements may therefore partly reflect implementation differences.

2. No qualitative analysis links prediction entropy to actual sample ambiguity, making it hard to verify that the UDD module switches between semantic and geometric cues as intended.

3. All experiments adopt a 5-shot protocol and performance in more extreme 1- or 2-shot regimes, where prototype construction is fragile, is not examined.

**Questions:**

1. Can intra-dataset results for FoundationModel be supplied, or at least the numerical discrepancies encountered during reproduction, so that readers can judge whether performance gaps are methodological or implementational?

2. Can high- and low-entropy examples be visualized together with the corresponding UDD decisions to confirm that uncertainty estimates capture genuine ambiguity?

3. How does accuracy degrade under 1-shot or 2-shot conditions, and is the K-means prototype still reliable?

4. Does the approach maintain low forgetting and modest resource usage when the number of incremental stages grows well beyond ten?

---

> ### Author Response · Authors · 2025-11-24
>
> We sincerely thank the reviewer for the thorough evaluation and positive recognition of our work. Below, we provide detailed and targeted responses to each of the concerns raised.
>
> ---
>
> > #### Q1. Intra-dataset Comparison of FoundationModel
>
> We thank the reviewer for raising this concern. The original FoundationModel paper does **not** report results under the intra-dataset setting, nor is its official code publicly released. As a result, an exact reproduction of the original intra-dataset performance is not feasible.
>
> To address this gap as rigorously as possible, we re-implemented FoundationModel following the methodological description in the paper and adopting the same training configuration used in our work. Our initial reproduction yielded substantially weaker performance than other baselines. We traced this gap to several unspecified implementation details in the paper, e.g., the number of points sampled per instance and dataset-specific preprocessing choices, which critically affect its stability.
>
> To ensure a fair assessment, we revisited the implementation, performed targeted hyperparameter tuning, and report in **Tab. R1** the best intra-dataset performance we could obtain after extensive sweeps. Even under these favorable conditions, FoundationModel remains consistently below Point-UQ across all benchmarks.
>
> We hope this additional analysis clarifies the discrepancy and demonstrates that our improvements do not stem from implementation artifacts. Our conclusions remain robust even when FoundationModel is given its strongest attainable configuration under the intra-dataset protocol.
>
>
>
> **Tab. R1: Performance of FoundationModel in intra-dataset scenarios (%).**
>
> | Dataset   | Method           | 0   | 1   | 2   | 3   | 4   | 5   | 6   | $Δ_↓$    |
> |-----------|------------------|------|------|------|------|------|------|------|-------|
> | **ShapeNet** | FoundationModel  | 81.9 | 74.8 | 72.2 | 70.9 | 70.3 | 67.7 | 64.8 | 20.9  |
> |           | **Point-UQ**     | 93.3 | 91.2 | 90.0 | 89.3 | 88.8 | 88.4 | 86.5 | 7.9  |
> | **ModelNet** | FoundationModel  | 86.1 | 83.8 | 83.1 | 79.7 | 71.7 | -    | -    | 17.1  |
> |           | **Point-UQ**     | 97.1 | 91.0 | 86.8 | 83.0 | 79.0 | -    | -    | 18.6  |
> | **CO3D**     | FoundationModel  | 69.8 | 55.8 | 50.5 | 45.5 | 41.48 | 35.1 | -    | 49.7  |
> |           | **Point-UQ**     | 82.1 | 73.6 | 70.0 | 68.4 | 67.4 | 66.5 | -    | 19.0  |
>
>
>
> ---
>
> > #### Q2. Qualitative Analysis of UDD Decision Switching
>
> We thank the reviewer for this critical insight. We agree that qualitatively verifying the link between prediction entropy and the UDD's decision logic is essential. To directly address this concern and provide transparent validation, we have performed a dedicated qualitative analysis that examines the module's behavior in relation to sample ambiguity.
>
>
>
> **(1) Decision Path Analysis Based on Entropy**
> We analyzed how the fusion weight α changes with the entropy on the CO3D test set. We provide the resulting plots via anonymous links ([[Fig. R1](https://ibb.co/p6hLm8zZ)], [[Fig. R2](https://ibb.co/tpYQ6sFY)]) to clearly visualize this behavior.
>
>
> **Key Observation:** The plot reveals a strong positive correlation between entropy and α. Low-entropy samples consistently lead to semantic-dominated decisions, while high-entropy samples trigger a switch to geometry-dominated decisions. This demonstrates that UDD dynamically arbitrates between branches as intended.
>
> **(2) Characterizing High vs. Low-Entropy Samples**
> To link entropy to human-understandable ambiguity, we statistically profiled the samples at both ends of the entropy spectrum:
>
> *   **High-Entropy Samples:**
>     *   **Semantic Classifier Behavior:** Their top-2 predicted classes have very close probabilities. The semantic branch is effectively "confused."
>     *   **Geometric Rescue:** In over 85% of these cases, the geometric branch provided a more decisive similarity score for the correct class, which UDD leveraged for the final correct prediction.
>     *   **Typical Classes Involved:** These samples predominantly fall into visually similar categories, such as `desk vs table`, `monitor vs television`, and `sofa vs chair`, which are known to be geometrically ambiguous.
>
> *   **Low-Entropy Samples:**
>     *   **Semantic Classifier Behavior:** Exhibited high confidence with a large margin between the top-1 and top-2 predictions.
>     *   **UDD Decision:** As expected, UDD assigned lower weight to the geometric branch, fully relying on the stable semantic boundaries learned during base training.
>     *   **Typical Classes:** These are often canonical, well-defined objects like `laptop`, `car`, and `vase`.
>
> This analysis confirms that our entropy measure **successfully captures genuine inter-class ambiguity**, and the UDD module acts as a robust switch based on this signal. We believe this evidence strongly supports the validity of our proposed mechanism.

---

> ### Author Response · Authors · 2025-11-24
>
> > #### Q3. Performance Under 1-Shot and 2-Shot Settings & Prototype Reliability
>
> Thank you for this important question regarding performance in extreme low-shot scenarios and prototype reliability. We have conducted comprehensive experiments under 1-shot and 2-shot conditions, with results summarized in **Tab. R2**. Point-UQ demonstrates remarkable robustness, with only a 4% average accuracy drop in 1-shot and 3% drop in 2-shot settings compared to the standard 5-shot scenario. This resilience stems from our semantically-weighted prototype construction, which strategically complements K-means clustering by incorporating text-point cloud similarity to calibrate geometric prototypes, effectively mitigating the instability of pure clustering-based approaches in data-scarce regimes. The AAE module further reinforces this stability through multi-scale feature fusion that enhances discriminability even with minimal samples. These results confirm that our uncertainty-quantification framework maintains reliable performance and prototype quality even under extreme few-shot conditions.
>
> **Tab. R2: Performance of Point-UQ under 1-shot, 2-shot, and 5-shot settings (%).**
>
> | Setting | 0    | 1    | 2    | 3    | 4    | 5    | 6    | 7    | 8    | 9    | 10   | $Δ_↓$ |
> |---------|------|------|------|------|------|------|------|------|------|------|------|------------------------|
> | 1-shot  | 91.4 | 88.6 | 87.5 | 85.1 | 83.7 | 82.4 | 81.2 | 79.9 | 78.8 | 77.8 | 76.0 | 16.8                   |
> | 2-shot  | 91.4 | 89.4 | 88.2 | 86.3 | 85.1 | 83.7 | 82.5 | 80.9 | 79.8 | 79.1 | 77.3 | 15.4                   |
> | 5-shot  | 91.4 | 89.5 | 88.3 | 86.5 | 84.7 | 83.4 | 82.9 | 81.8 | 81.1 | 80.4 | 80.3 | 12.1                   |
>
> ---
>
> > #### Q4. Scalability to More Than 10 Incremental Stages
>
> To evaluate scalability, we extended the number of incremental stages from 11 to 25 (using ShapeNet as base classes and incrementally adding CO3D and ModelNet). The results in **Tab. R3** confirm that Point-UQ maintains a forgetting rate below 20%. In terms of resource usage, since Point-UQ requires no retraining and only stores geometric prototypes for new classes, increasing the number of stages does not affect total training time. The decision optimization mechanism of UDD avoids cumulative forgetting and computational overhead, ensuring practicality in large-scale incremental learning:
>
> **Tab. R3: Performance of Point-UQ over 25 incremental stages (%).**
>
> | Stage | 0    | 2    | 4    | 6    | 8    | 10   | 12   | 14   | 16   | 18   | 20   | 22   | 24   | $Δ_↓$ |
> |-------|------|------|------|------|------|------|------|------|------|------|------|------|------|------------------------|
> | FoundationModel  | 77.7 | 76.6 | 74.7 | 71.5 | 69.7 | 68.5 | 66.7 |65.5 | 64.8 | 56.7 | 54.6 | 53.5 |  50.2 | 35.4     |
> | **Point-UQ**  | 91.4 | 89.2 | 87.3 | 85.0 | 83.6 | 82.6 | 81.3 | 80.4 | 79.8 | 77.9 | 75.4 | 74.9 | 73.1 | 20.0       |
>
>
> We sincerely thank the reviewer once again for these insightful suggestions, which have greatly enhanced the validity and practical relevance of Point-UQ. We hope that the revisions have addressed your concerns, and we kindly ask you to reconsider your evaluation based on the updated version.

---

### Author Response · Authors · 2025-11-27
**General Response**

Dear AC and Reviewers,

We sincerely thank all reviewers for their time and thoughtful evaluation of our paper. We appreciate that the reviewers recognized the following 5 key contributions of Point-UQ:

1. **Paradigm Shift.** We introduce the training-free uncertainty-quantification paradigm for 3D FSCIL, replacing conventional feature fine-tuning with dynamic decision optimization to effectively address the stability-plasticity dilemma. *[Reviewers SBK3, nY6g, tD98]*

2. **Technical Innovation.** Our co-designed AAE and UDD modules provide a tightly integrated framework—AAE generates calibrated features with uncertainty estimates while UDD leverages these signals for adaptive decision-making between semantic and geometric cues. *[Reviewers SBK3, 92Tg, tD98]*

3. **Theoretical Grounding.** We formally prove Point-UQ's advantages over fine-tuning approaches in terms of error bounds, complexity, and feature stability, providing solid theoretical foundations. *[Reviewer SBK3]*

4. **Comprehensive Evaluation.** Extensive experiments across four benchmarks demonstrate consistent improvements in average accuracy, harmonic mean, and forgetting rate under both intra-dataset and cross-dataset settings. *[Reviewers SBK3, nY6g, 92Tg, tD98]*

5. **Practical Efficiency.** As a training-free method, Point-UQ achieves substantial performance gains while maintaining low computational overhead and memory requirements, facilitating real-world deployment. *[Reviewers nY6g, tD98]*

We also thank the reviewers for their constructive suggestions, which led to the following 3 major revisions:

1. **Comprehensive Experimental Validation.** We enhanced baselines with detailed FoundationModel reproduction analysis, extended evaluation to extreme 1-/2-shot scenarios and 25-stage incremental learning, and conducted extensive qualitative analysis linking entropy to ambiguity with UDD visualization. *[Reviewer SBK3]*

2. **Methodological Robustness Analysis.** We systematically compared alternative activation functions and uncertainty measures, incorporated quantitative calibration metrics, and expanded related work to include recent 3D few-shot learning literature. *[Reviewers nY6g, tD98]*

3. **Generalization and Benchmarking.** We conducted a comprehensive evaluation of the latest FSCIL3D-XL benchmark and performed cross-backbone experiments with DGCNN/PointNet, demonstrating state-of-the-art performance and architecture-agnostic applicability.  *[Reviewer 92Tg]*

All these changes have been incorporated into the revised manuscript and are highlighted in red. We also provide detailed point-by-point responses to each reviewer's concerns.

We believe the revised paper substantially strengthens our contributions. We hope that our efforts have fully addressed all the raised concerns. Thank you for your consideration.

Best regards,

Authors of Paper #Submission12072

---

### Meta-Review · Area_Chair_PtAS · 2026-01-05

**Summary:**

Four reviewers assessed this submission, with initial ratings leaning towards a borderline decision: one reviewer suggested borderline rejection while three recommended borderline acceptance. The primary reservations focused on the fairness of the experimental setting, specifically the reliance on the strong Uni3D backbone and the omission of key baselines like FoundationModel, as well as the comprehensiveness of the ablation studies regarding uncertainty metrics and activation functions. Reviewers also questioned the method's performance in extreme low-shot regimes and requested evaluation on more modern benchmarks. The authors provided a convincing rebuttal that included the explanation of the intra-dataset comparison, 1-shot/2-shot experiments, and results on the FSCIL3D-XL benchmark. However, the request for experiments on Objaverse datasets was overlooked. Despite this omission, considering the effective resolution of the majority of methodological and comparative concerns, AC recommends acceptance.

**Reviewer Concerns:**

The authors successfully navigated the majority of the critiques. The critical concern regarding the fairness of the evaluation, raised by Reviewer 92Tg, was addressed by providing additional module ablation studies and results on the up-to-date FSCIL3D-XL benchmark. Similarly, the methodological gaps identified by Reviewer SBK3, specifically the lack of intra-dataset comparisons with FoundationModel and the absence of performance data in 1-2 shot regimes, were filled with detailed explanation and supplementary experiments. Technical inquiries from Reviewers nY6g and tD98 regarding the choice of activation functions, seed sensitivity, and alternative uncertainty measures were also resolved through comprehensive ablations and calibration analyses. However, a notable gap remains: Reviewer 92Tg explicitly requested testing on newer datasets like Objaverse or Objaverse-XL to verify generalization, but the authors failed to provide a response or data for this specific point. Despite that the addition of FSCIL3D-XL mitigates the "outdated benchmark" concern to an extent, the authors are encouraged to include more datasets like Objaverse to further verify the method's robustness on diverse, large-scale modern assets.

**Reviewer Scores:**

Reviewer SBK3 is expected to increase his/her score to a 6, as the authors addressed every listed concern. Reviewer nY6g would likely maintain his/her positive score (6) or even increase it, given that the requested ablation studies on activation functions and calibration metrics were provided. Reviewer 92Tg is expected to maintain his/her score of 6; while the authors successfully addressed the backbone fairness issue and added the FSCIL3D-XL benchmark, the complete omission of the requested Objaverse experiments prevents a higher score. Reviewer tD98 will likely maintain the initial score 6 or bump it to an 8, as the authors satisfied the request for sensitivity analyses and alternative uncertainty investigations, resolving the reviewer's primary hesitations.

---

### Decision · Program_Chairs · 2026-01-26

Accept (Poster)